# Global tropospheric hydroxyl distribution, budget and reactivity

Jos Lelieveld[1], Sergey Gromov[1], Andrea Pozzer[1], and Domenico Taraborrelli[1]

[1]Max Planck Institute for Chemistry, Atmospheric Chemistry Department, P.O. Box 3060, 55020 Mainz, Germany

*Correspondence to*: Jos Lelieveld (jos.lelieveld@mpic.de)

**Abstract.** The self-cleaning or oxidation capacity of the atmosphere is principally controlled by hydroxyl (OH) radicals in the troposphere. Hydroxyl has primary ($P$) and secondary ($S$) sources, the former mainly through the photo-dissociation of ozone, the latter through OH recycling in radical reaction chains. We used the recent Mainz Organics Mechanism (MOM) to advance volatile organic carbon (VOC) chemistry in the general circulation model EMAC, and show that $S$ is larger than previously assumed. By including emissions of a large number of primary VOC, and accounting for their complete

breakdown and intermediate products, MOM is mass conserving, and calculates substantially higher OH reactivity from VOC oxidation compared to predecessor models. Whereas previously $P$ and $S$ were found to be of similar magnitude, the present work indicates that $S$ may be twice as large, mostly due to OH recycling in the free troposphere. Further, we find that nighttime OH formation may be significant in the polluted subtropical boundary layer in summer. With a mean OH recycling probability of about 67%, global OH is buffered and not sensitive to perturbations by natural or anthropogenic emission

changes. Complementary primary and secondary OH formation mechanisms in pristine and polluted environments in the continental and marine troposphere, connected through long-range transport of $O_3$, can maintain stable global OH levels.

## 1 Introduction

The removal of most natural and anthropogenic gases from the atmosphere, important for air quality, the ozone layer and climate, takes place through their oxidation by hydroxyl (OH) radicals in the troposphere. The central role of tropospheric

OH in the atmospheric oxidation capacity (or efficiency) has been recognized since the early 1970s (Levy, 1971; Crutzen, 1973; Logan et al., 1981, Ehhalt et al., 1991). The primary OH formation rate ($P$) depends on the photo-dissociation of ozone ($O_3$) by ultraviolet (UV) sunlight – with a wavelength of the photon ($h\nu$) shorter than 330 nm – in the presence of water vapor

$$O_3 + h\nu \ (\lambda < 330 \ \text{nm}) \ \rightarrow \ O(^1D) + O_2 \tag{R1}$$

$$O(^1D) + H_2O \ \rightarrow \ 2OH \tag{R2}$$

(Note that the formal notation of hydroxyl is HO·, indicating one unpaired electron on the oxygen atom. For brevity we omit the dot and use the notation OH, and similarly for other radicals.) Since the stratospheric ozone layer in the tropics is relatively thin, UV radiation is less strongly attenuated compared to the extra-tropics, and also because the solar zenith angle

and water vapor concentrations are relatively high, zonal OH is highest at low latitudes in the lower to middle troposphere (Crutzen and Zimmermann, 1991; Spivakovsky et al., 2000).

The OH radicals attack reduced and partly oxidized gases such as methane ($CH_4$), non-methane volatile organic compounds (VOCs) and carbon monoxide (CO), so that these gases only occur in trace amounts, e.g.,

$\qquad$ $CO + OH \qquad \rightarrow CO_2 + H$ $\hfill$ (R3)

$\qquad$ $H + O_2 (+M) \qquad \rightarrow HO_2 (+M)$ $\hfill$ (R4)

where M is an air molecule that removes excess energy from reaction intermediates by collisional dissipation. Because OH is highly reactive it has an average tropospheric lifetime of about 1-2 seconds. After the initial OH reaction (R3) peroxy radicals are produced (R4), which can combine to form peroxides

$\qquad$ $HO_2 + HO_2 \qquad \rightarrow H_2O_2 + O_2$ $\hfill$ (R5)

$\qquad$ $RO_2 + HO_2 \qquad \rightarrow ROOH + O_2$ $\hfill$ (R6a)

$\qquad$ $\rightarrow RO + O_2 + OH$ $\hfill$ (R6b)

$\qquad$ $\rightarrow ROH + O_3$ $\hfill$ (R6c)

RH is a VOC from which OH can abstract the hydrogen to form water and an alkyl radical, which reacts with $O_2$ to form a

peroxy radical, $RO_2$. After peroxide formation (R5, R6a) the reaction chains can either propagate or terminate, the latter by deposition. Propagation of the chain leads to higher generation reaction products and secondary OH formation ($S$), which can be understood as OH recycling. For example, the photolysis of ROOH leads to OH production. In air that is directly influenced by pollution emissions $S$ is largely controlled by nitrogen oxides ($NO+NO_2=NO_X$)

$\qquad$ $NO + HO_2 \qquad \rightarrow NO_2 + OH$ $\hfill$ (R7)

This reaction, referred to as the $NO_X$ recycling mechanism of OH, also leads to ozone production through photo-dissociation of $NO_2$ by ultraviolet and visible light

$\qquad$ $NO_2 + hv\ (\lambda<430nm) \rightarrow NO + O(^3P)$ $\hfill$ (R8)

$\qquad$ $O(^3P) + O_2 (+M) \qquad \rightarrow O_3 (+M)$ $\hfill$ (R9)

However, in strongly polluted air $NO_2$ can locally be a large OH sink, and in such environments the net effect of $NO_X$ on OH

is self-limiting through the reaction

$\qquad$ $NO_2 + OH (+M) \qquad \rightarrow HNO_3 (+M)$ $\hfill$ (R10)

Conversely, under low-$NO_X$ conditions, mostly in pristine air, secondary OH formation by other mechanisms is important

$\qquad$ $O_3 + HO_2 \qquad \rightarrow 2O_2 + OH$ $\hfill$ (R11)

$\qquad$ $H_2O_2 + hv\ (\lambda<550\ nm) \rightarrow OH + OH$ $\hfill$ (R12)

These reactions are referred to as the $O_X$ recycling mechanism of OH. In prior work we suggested that the strong growth of air pollution since industrialization, especially in the 20[th] century, has drastically changed OH production and loss rates, but that globally the balance between $P$ and $S$ changed little (Lelieveld et al., 2002). This is associated with a relatively constant OH recycling probability $r$, defined as $r = 1 - P/G$, in which $G$ is gross OH formation ($G=P+S$); $P$, $S$ and $G$ have unit

moles/year. We computed that globally $r$ changed little since pre-industrial times, remaining at about 50%. Thus, in the past century $G$ (the atmospheric oxidation power) kept pace with the growing OH sink related to the emissions of reduced and partly oxidized pollution gases. Lelieveld et al. (2002) performed perturbation simulations, applying pulse emissions of $NO_X$ and $CH_4$, to compute the impact on OH. This showed that at an OH recycling probability of 60% or higher, these perturbations have negligible influence on OH (their Fig. 6). Therefore, at $r>60\%$ the atmospheric chemical system can be considered to be buffered.

While globally $r$ has remained approximately constant, the mean tropospheric OH concentration and the lifetime of $CH_4$ ($\tau_{CH4}$) have also changed comparatively little, for example within a spread of about 15% calculated by a 17-member ensemble of atmospheric chemistry-transport models (Naik et al., 2013). Despite substantial differences in OH concentrations and $\tau_{CH4}$ among the models, simulations of emission scenarios according to several Representative Concentration Pathways indicate that future OH changes will probably also be small, i.e., well within 10% (Voulgarakis et al., 2013). We interpret the relative constancy of $r$, mean OH and $\tau_{CH4}$ as indication that global OH is buffered against perturbations. This is corroborated by studies based on observations of methyl chloroform, with known sources and OH reaction as the main sink, showing small inter-annual variability of global OH and small inter-hemispheric difference in OH (Krol and Lelieveld, 2003; Montzka et al., 2011; Patra et al., 2014).

For our previous estimates of $P$ and $S$ we used a chemistry-transport model with the Carbon Bond Mechanism (CBM) to represent non-hydrocarbon chemistry (Houweling et al., 1998). This mechanism aggregates organic compounds into categories of species according to molecular groups, and has been successfully used to simulate ozone concentrations with air quality models (Stockwell et al., 2012). However, such chemical schemes are not mass-conserving, e.g., for carbon, and are optimized for conditions in which $NO_X$ dominates $r$, while in low-$NO_X$ environments other mechanisms may be important, for example through the chemistry of non-methane VOCs emitted by vegetation (Lelieveld et al., 2008), as reviewed by Vereecken and Francisco (2012), Stone et al. (2012) and Monks et al. (2015). A limitation of the CBM and other, similar mechanisms is that 2nd and higher generation reaction products are lumped or ignored for computational efficiency, whereas they can contribute importantly to OH recycling and ozone chemistry (Butler et al., 2011; Taraborrelli et al., 2012).

Here we apply the Mainz Organics Mechanism (MOM) that accounts for recent developments in atmospheric VOC chemistry (Taraborrelli et al., manuscript in preparation). The MOM is a further development of the Mainz Isoprene Mechanism (Taraborrelli et al., 2009, 2012). In addition to isoprene, MOM computes the chemistry of saturated and unsaturated hydrocarbons, including terpenes and aromatics (Cabrera-Perez et al., 2016). We use it to estimate the role of radical production through reactions of oxidized VOC, referred to as OVOC recycling mechanism of OH, being contrasted with the $NO_X$ and $O_X$ recycling mechanisms of OH. Based on this scheme, implemented in the atmospheric chemistry – general circulation model EMAC, we provide an update of global OH calculations, sources, sinks, tropospheric distributions, OH reactivity, the lifetime of $CH_4$ and CO, and discuss implications for atmospheric chemistry. We contrast the boundary layer and free troposphere (BL and FT), the Northern and Southern Hemisphere (NH and SH) and the tropics and extra-

tropics. We show that complementary OH recycling mechanisms in terrestrial, marine, pristine and polluted environments, inter-connected through atmospheric transport, sustain stable levels of hydroxyl in the global troposphere.

## 2 VOC chemistry and model description

To reconcile observations of high OH concentrations over the Amazon rainforest with models that predicted low OH
concentrations, we have proposed that the chemistry of isoprene recycles OH, e.g., involving organic peroxy radicals (Lelieveld et al., 2008). Progress on such reactions was reported by Taraborrelli et al. (2012) and incorporated in a predecessor version of the present chemistry scheme. Laboratory experimental results by Groß et al. (2014a,b) provided additional evidence and insight into this type of chemistry, indicating that OH formation via reaction R6b ($RO_2+HO_2$) had previously been underestimated significantly. While in polluted air peroxy radicals preferentially react with NO, in pristine,
low-$NO_X$ conditions over the rain forest, e.g., in the Amazon, isoprene degradation leads to hydroxy-hydroperoxides, which can reform OH upon further oxidation (Paulot et al., 2009).

     An important pathway in isoprene chemistry, basic to the recycling of OH, is isomerization through H-migration within oxygenated reaction products, leading to photo-labile hydroperoxy-aldehydes (HPALD), as reviewed by Vereecken and Francisco (2012). While a high rate of 1,5-H-shifts that we have assumed previously (Taraborrelli et al., 2012) was not
confirmed experimentally, these and especially 1,4-H- and 1,6-H-shifts have nevertheless shown to be key intermediaries in OH recycling (Crounse et al., 2012, 2013; Fuchs et al., 2014; Peeters et al., 2014). When the OH concentration is low, its formation is maintained by photo-dissociation of HPALD, while at high OH concentration its sink reaction with HPALD gains importance. Next to HPALD, unsaturated hydroperoxyaldehydes, e.g., peroxyacylaldehydes (PACALD), were shown to be relevant (Peeters et al., 2014). Higher generation reaction products include several organic peroxides that produce OH
upon photo-dissociation, which need to be accounted for in atmospheric chemistry models to reproduce field and reaction chamber observations (Nölscher et al., 2014).

     These reactions have been included into the Mainz Organics Mechanism (MOM), being an extension and update of the Mainz Isoprene Mechanism, v.2 (Taraborrelli et al., 2009, 2012). The scheme, which accounts for about 630 compounds and 1630 reactions, makes use of rate constant estimation methods similarly to the Master Chemical Mechanism by Jenkin et al.
(2015) (http://mcm.leeds.ac.uk/MCM), and recommendations by the Task Group on Atmospheric Chemical Kinetic Data Evaluation (http://iupac.pole-ether.fr), in addition to our own evaluation of recent literature. For the present work we applied the full scheme, also used by Cabrera-Perez et al. (2016), which is computationally demanding and precludes that we apply high spatial resolution of the model for extended time periods. For computational efficiency in global and regional models, the scheme will be condensed in future (Taraborrelli et al., in preparation). In contrast to some previous chemistry
mechanisms in atmospheric models, MOM accounts for higher generation reaction products and is mass conserving (notably for carbon containing reaction products from VOC oxidation).

The MOM has been included into the ECHAM/MESSy Atmospheric Chemistry (EMAC) general circulation model. The core atmospheric general circulation model is ECHAM5 (Roeckner et al., 2006), coupled with the Modular Earth Sub-model System, of which we have applied MESSy2 version 2.42 (Jöckel et al., 2010). For this study EMAC was used in a chemical-transport model (CTM mode) (Deckert et al. 2011), i.e., by disabling feedbacks between photochemistry and dynamics. EMAC sub-models represent tropospheric and stratospheric processes and their interaction with oceans, land and human influences, and describe emissions, radiative processes, atmospheric multiphase chemistry, aerosol and deposition mechanisms (Jöckel et al., 2005, 2006; Sander et al., 2005, 2011, 2014; Kerkweg et al., 2006; Tost et al., 2006, 2007a; Pozzer et al., 2007, 2011; Pringle et al., 2010). We applied the EMAC model at T42/L31 spatial resolution, i.e., at a spherical spectral truncation of T42 and a quadratic Gaussian grid spacing of about 2.8 degrees latitude and longitude, and 31 hybrid terrain following – pressure levels up to 10 hPa.

Results have been evaluated against observations (Pozzer et al., 2010, 2012; de Meij et al., 2012; Christoudias and Lelieveld, 2013; Elshorbany et al., 2014; Yoon and Pozzer, 2014; Cabrera-Perez et al., 2016; for additional references, see http://www.messy-interface.org). Here we present results based on emission fluxes and meteorology representative of the year 2013, mostly annual means unless specifically mentioned otherwise. Tests of the present model version indicate minor changes, e.g., in intermediately long-lived compounds such as $O_3$ and CO, compared to previous versions. $N_2O$ and $CH_4$ concentrations have been prescribed at the surface based on observations. Anthropogenic emissions have been based on the RCP8.5 emission scenario (Riahi et al., 2007; van Vuuren et al., 2011; Meinshausen et al., 2011). The scenario was tested by Granier et al. (2011), indicating that it realistically represents the source strengths of pollutants after the year 2000. The RCP8.5 scenario was also applied by Cabrera-Perez et al. (2016), in which the emissions and chemistry of aromatic compounds has been described. Natural emissions of higher VOCs are interactively calculated, amounting to 760 TgC/yr, with a four year range of 747–789 TgC/yr (including about 73%, or 546–578 TgC/yr, of isoprene) (Guenther et al., 2012), and anthropogenic emissions of saturated, unsaturated and aromatic compounds amount to 105 TgC/yr. These flux integrals are in carbon equivalent. It should be mentioned that in previous generation atmospheric chemistry-transport models VOC emissions have been artificially reduced to prevent the collapse of OH concentrations in regions of strong natural sources, i.e., at high-VOC and low-$NO_X$ conditions (Arneth et al., 2010).

To analyze model production and sink pathways of OH and $HO_2$, including multiple radical recycling, and compute fluxes of reactants and intermediate products, we used the kinetic chemistry tagging technique of Gromov et al. (2010). The scheme computes detailed turnover rates of selected tracers, in this case OH, $HO_2$, $O_3$, CO, aldehydes, peroxides and others, in various parts of the MOM chemistry scheme within EMAC. With limited additional computational load the extensive budgeting allows characterization of OH sources and sinks, while the diagnostic calculations are decoupled from the regular chemistry scheme.

Here we present a selection of results, focusing on annual and large-scale averages to characterize global OH. The Supplement presents supporting tables and figures for the interested reader. Pages S1-S14 illustrate time sequences (Hovmöller plots), seasonal differences and results for different atmospheric environments and reservoirs such as the

boundary layer (BL) and free troposphere (FT), to distinguish continental from marine boundary layers (CBL and MBL), the lower troposphere from the tropopause region and the lower stratosphere. These supplementary results focus on distributions of OH and $HO_2$, and lifetimes of different species, notably OH, $HO_2$, CO and $CH_4$, and include figures of global OH reactivity which are relevant for the discussion in Sect. 5. Page S15 presents details on the global OH budget, relevant for Sect. 6. The Supplement also includes scatter plots between observations and model results of CO and $O_3$ at the surface for the year 2013 (S16), a table with details of VOC emission fluxes applied in EMAC (S17), and the complete mechanism of MOM, including a list of all chemical species (S18 and following). Model calculated global datasets of OH concentrations and other trace gases are available upon request.

**3 Global OH distribution**

In agreement with previous studies our model calculations show highest OH concentrations in the tropical troposphere (Fig. 1). Globally, mean tropospheric OH is $11.3\cdot10^5$ molecules/$cm^3$, close to the multi-model mean of $11.1\pm1.6\cdot10^5$ molecules/$cm^3$ derived by Naik et al. (2013) for the year 2000. Note that these are volume weighted means. Following the recommendation by Lawrence et al. (2001) we also calculated the air mass weighted ($11.1\cdot10^5$ molecules/$cm^3$), $CH_4$ weighted ($12.4\cdot10^5$ molecules/$cm^3$) and methyl chloroform (MCF) weighted means ($12.3\cdot10^5$ molecules/$cm^3$), though henceforth primarily report volume weighted mean values.

The calculated tropical tropospheric average is $14.6\cdot10^5$ molecules/$cm^3$ (between Tropics of Cancer and Capricorn), with the NH and SH extra-tropical averages being 9.1 and $6.6\cdot10^5$ molecules/$cm^3$, respectively. Our model indicates more OH north of the Equator compared to the south, $12.1\cdot10^5$ and $10.1\cdot10^5$ molecules/$cm^3$, respectively. Hence the NH/SH ratio is 1.20, being towards the low end of the multi-model estimate of $1.28\pm0.10$ by Naik et al. (2013), though deviating from inter-hemispheric parity derived by Patra et al. (2014) based on the analysis of MCF ($CH_3CCl_3$) measurements.

For the air mass, $CH_4$ and MCF weighted means we find NH/SH ratios of 1.25, 1.30 and 1.25, respectively. Part of the discrepancy with Patra et al. (2014) may be related to the seasonally varying position of the Inter-Tropical Convergence Zone (ITCZ), which effectively separates the meteorological NH from the SH. The position of the ITCZ, on average a few degrees north of the equator in the region of highest OH, can influence these calculations, both in models and MCF analyses. If we correct for this, the volume weighted NH/SH ratio of OH decreases from 1.20 to 1.13. In the extra-tropics our model calculates 28% less OH in the SH compared to the NH, being the main reason for the model calculated inter-hemispheric OH disparity. The difference is even larger between the Arctic and Antarctic regions (defined by the polar circles), as the calculated mean OH concentration is 50% lower in the latter. However, if we also include the lower stratosphere (up to 10 hPa) we find near-interhemispheric parity of OH, i.e., 5% more in the NH and only 2% more based on the ITCZ metric. Considering the importance of the stratosphere as an MCF reservoir to the troposphere in recent years (Krol and Lelieveld, 2003), and possible inter-hemispheric differences in the age-of-air in the middle atmosphere, these aspects should be

investigated further with a model version that accounts for the atmosphere from the surface to the mesosphere, to investigate the importance for MCF analyses and inferred OH distributions.

Fig. 1 illustrates that high OH concentrations in the tropics can extend up to the tropopause, with a main OH maximum below 300-400 hPa and a second maximum between 200 and 150 hPa. Note that the tropopause in the tropics is defined using temperature and in the extra-tropics potential vorticity gradients (2 PV units). The oxidative conditions throughout the tropical troposphere limit the flux of reduced and partly oxidized gases (e.g., reactive halocarbons, sulfur and nitrogen gases) into the stratosphere through their chemical conversion into products that are removed by deposition processes. Near the cold tropical tropopause reaction products, such as low-volatile acids, can be removed by adsorption to sedimenting ice particles that also dehydrate the air that ascends into the stratosphere (Lelieveld et al., 2007). Due to the slow ascent rates of air parcels in the tropical tropopause region (tropical transition layer), pollutant gases are extensively exposed to oxidation by OH for several weeks to months. This mechanism protects the ozone layer from $O_3$ depleting substances that could be transported from the troposphere, at least to the extent that they react with OH.

In the global troposphere annual column average OH ranges from $1.0 \cdot 10^5$ to $22.0 \cdot 10^5$ molecules/cm$^3$, i.e., between high and low latitudes, respectively (Fig. 1). This range is determined by the meridional OH gradient in the FT, since about 85% of tropospheric OH formation takes place in the FT, which dominates the global OH distribution (detailed below). In the BL the range is much larger, $0.3 \cdot 10^5$ to $44.0 \cdot 10^5$ molecules/cm$^3$, as OH is affected by variable surface emissions. The subordinate role of the BL in the global OH load and distribution is conspicuous, for example from the OH maximum in the BL over the Middle East and OH minima over the Central African and Amazon forests (Fig. S1 of Supplement), which do not appear in the tropospheric column average OH concentrations, as the latter follow the OH distribution in the FT (Fig. 1).

In the BL over tropical forests OH concentrations are comparatively low, about $10 \cdot 10^5$ to $20 \cdot 10^5$ molecules/cm$^3$, in agreement with OH measurements in South America and Southeast Asia (Kubistin et al., 2010; Pugh et al., 2010; Whalley et al., 2011), while in the FT in these regions OH concentrations are several times higher. The relatively high OH in the tropical FT is related to the combination of emissions from vegetation with NO$_X$ from lightning in deep thunderstorm clouds. This is most prominent over Central Africa where deep convection and lightning are relatively intense (Fig. 1, left panel). The latter was corroborated by comparing our model with lightning observations (Tost et al., 2007b). The chemical mechanisms that control OH in the BL and FT are connected through vertical transport and mixing, which balances formation and loss in the column, i.e., near the surface VOCs are a net sink of OH while their reaction products are a net OH source aloft.

In the NH extra-tropics mean OH in the MBL approximately equals that in the CBL, i.e., in the zonal direction. As shown previously, this is related to the transport and mixing of oxidants (primarily $O_3$) and precursor gases (e.g., NO$_X$ and partially oxidized volatile organic compounds, OVOCs) from polluted regions across the Atlantic and Pacific Oceans (Lelieveld et al., 2002). In the SH, on the other hand, where anthropogenic NO$_X$ sources and related transports are much weaker, mean OH in the CBL is about 15% higher compared to the MBL. In the extra-tropical troposphere as a whole, OH gradients in the longitudinal direction are typically small (Fig. 1), related to relatively rapid exchanges by zonal winds in transient synoptic weather systems.

While primary OH formation (R1,2) during daytime is controlled by photo-dissociation of $O_3$, there are additional sources that can be relevant at night. This includes reactions of $O_3$ with unsaturated hydrocarbons and aromatic compounds in polluted air and with terpenes emitted by vegetation. Fig. 2 shows nighttime OH in the boundary layer during January and July to illustrate the strong seasonal dependency. While the color coding is the same as Fig. 1, the concentrations are scaled by a factor 20. On a global scale, OH concentrations in the BL at night are nearly two orders of magnitude lower than during the day, and in the FT diel differences are even larger. Therefore, nighttime OH does not significantly influence the atmospheric oxidation capacity and the lifetimes of $CH_4$ and CO. Nevertheless, Fig. 2 shows several hotspots, mostly in the subtropical BL in the NH during summer, where nighttime OH can exceed $10^5$ molecules/cm$^3$ and could contribute to chemical processes including new particle formation. These regions include the Western USA, the Mediterranean and Middle East, the Indo-Gangetic Plain and Eastern China.

## 4 Global HO$_X$ distribution

Since conversions between $HO_2$ and OH play a key role in OH recycling, we address the budget of HO$_X$ (OH+$HO_2$), which is dominated by $HO_2$. Field and laboratory measurements often address both OH and $HO_2$. Fig. 3 shows the annual $HO_2$ concentration distribution, the counterpart of OH in Fig. 1. We find that in the BL annual mean $HO_2$ ranges from 0.1 to 6.4·$10^8$ molecules/cm$^3$ globally, whereas in the FT as a whole this is only 0.2 to 1.1·$10^8$ molecules/cm$^3$. Even though the mean lifetime of $HO_2$ in the troposphere of 1.5 minutes is much longer than of OH (factor 60), both OH and $HO_2$ are locally controlled by chemistry. Transport processes influence HO$_X$ through longer-lived precursor and reservoir species such as $O_3$ and OVOCs. Whereas OH in the BL over the tropical forests is relatively low, $HO_2$ is relatively high, about 5·$10^8$ molecules/cm$^3$, i.e., 2 to 3 orders of magnitude higher than OH, consistent with observations (Kubistin et al., 2010). Our results suggest that from a global perspective HO$_X$ is highest over the tropical forests, where photochemistry is very active and OH sources and sinks are large. Localized HO$_X$ maxima are also found in the polluted CBL where reactive VOC and NO$_X$ emissions are strong, e.g., by the petroleum industry north of the Mexican Gulf and near the Persian Gulf (Ren et al., 2013; Lelieveld et al., 2009).

On a global scale the tropospheric production of HO$_X$ is dominated by that in the FT. In the FT HO$_X$ is subject to long-range transport of relatively long-lived source and sink gases such as $O_3$ and CO, whereby the latter redistributes OH into $HO_2$ within HO$_X$, whereas in the BL local emissions of short-lived VOCs and NO$_X$ are more relevant. The efficient atmospheric transport of longer-lived gases, such as $O_3$ from both the stratosphere and photochemically polluted regions, helps buffer the OH formation in regions where oxidant is depleted, such as the MBL (Lelieveld and Dentener, 2000; de Laat and Lelieveld, 2000). Within the tropospheric column, convection and entrainment of $O_3$ rich air from the FT into the BL play a key role in the exchange of oxidant, which reduces vertical gradients, and balances HO$_X$ production and loss processes across altitudes.

We calculate a global tropospheric average $HO_2$ concentration of $0.6 \cdot 10^8$ molecules/cm³. We find roughly the same average concentrations in the tropical and NH extra-tropical troposphere, and slightly less in the SH extra-tropics ($0.5 \cdot 10^8$ molecules/cm³). Thus the mean tropospheric $HO_2$ (and $HO_X$) concentrations in these tropical and extra-tropical reservoirs are very similar. Nevertheless, in the SH the mean $HO_2$ concentration in the CBL is about a factor 2 higher compared to the
MBL, associated with strong VOC emissions by vegetation subject to intense photochemistry. In the NH mean $HO_2$ is comparable between the MBL and CBL, due to the widespread impact of air pollution, as explained above. The seasonal differences in tropospheric $HO_X$ at middle and high latitudes can be large though, i.e., about an order of magnitude between summer and winter. The seasonality of primary OH formation, which is proportional to solar radiation intensity, is even larger. In Sec. 6 we discuss that the low primary formation in winter is partly compensated by secondary OH formation,
being less dependent on sunlight, which reduces latitudinal and seasonal OH contrasts.

## 5 Trace gas lifetimes and OH reactivity

The average tropospheric lifetime of OH ($\tau_{OH}$) is 1.5s, calculated by dividing the annual averages of the volume-weighted OH burden and the total photochemical sink rate. Fig. 4 presents the spatial distribution of $\tau_{OH}$. Unlike the OH concentration, $\tau_{OH}$ does not exhibit a strong seasonal cycle, being nearly absent in the tropics and the FT. Only in the CBL over Siberia,
around 60°N, seasonal differences can reach a factor 5, related to the annual variability of VOC emissions by boreal forest (Siberian taiga). The tropospheric mean $\tau_{OH}$ in the NH is 1.4s and in the SH 1.6s. In the MBL mean $\tau_{OH}$ is about 0.7s, in the CBL about 0.3s. The longest $\tau_{OH}$ is found near the tropical tropopause (10-20s) where OH reactivity (the inverse of $\tau_{OH}$) is thus below 0.1 $s^{-1}$. While this is largely related to low temperatures and reduced reaction rates, it also indicates that air masses that traverse the tropical transition layer into the stratosphere are cleansed from reactive compounds that are removed
by OH, which is important for organo-halogen compounds, for example, that could damage the ozone layer. In the NH mean tropospheric OH reactivity is 0.7 $s^{-1}$, and in the SH 0.6 $s^{-1}$. The seasonality of $\tau_{HO2}$ is more pronounced than of $\tau_{OH}$; $\tau_{HO2}$ is longest in the cold season and over Antarctica, up to 10 minutes. In the MBL $\tau_{HO2}$ is on average 1.3 minutes, in the CBL 0.5 and in the FT 1.7 minutes.

We find that $\tau_{OH}$ is generally shortest over the tropical forest, followed by the boreal forest, coincident with the spatial
distribution of total OH reactivity, i.e., the inverse of $\tau_{OH}$, shown in Fig. 5. Near the Earth's surface the OH reactivity varies from about 0.5 $s^{-1}$ over Antarctica, due to reaction of OH with $CH_4$ and CO in clean and cold air, to approximately 100 $s^{-1}$ over the Amazon rainforest in the dry season due to relatively strong isoprene sources, complemented by biomass burning emissions. This modeled OH reactivity range seems realistic in comparison to observations, whereas previous models – as well as measurement techniques – that did not account for all VOC reaction products and intermediates, strongly
underestimated OH reactivity, i.e., up to a factor of ten (Walley et al., 2011; Mogensen et al., 2015; Nölscher et al., 2016). This topic will be studied in greater detail in a follow-up publication where we address the reactive carbon budget in

different environments, evaluated against measurements, where we also include secondary organic aerosols as described by Tsimpidi et al. (2016).

Our estimate of the mean lifetime of $CH_4$ due to oxidation by tropospheric OH ($\tau_{CH4}$) is 8.5 years, which is within the multi-model calculated $1\sigma$ standard deviation of the mean of 9.7±1.5 years presented by Naik et al. (2013), though towards the lower end of the range. Notice that this figure does not include uptake of $CH_4$ by soils and stratospheric loss by OH, $O(^1D)$ and chlorine radicals, which together make up about 10% of the total $CH_4$ sink. The 17 models that participated in the model inter-comparison by Naik et al. (2013) show a range of 7.1 – 14.0 years, while the multi-model mean of 9.7 years was considered to be 5-10% higher than observation-derived estimates.

One reason for our $\tau_{CH4}$ estimate being toward the lower end of the range may be that Naik et al. (2013) refer to the year 2000, whereas we applied an emission inventory for the year 2010, i.e., after a period when $NO_X$ concentrations increased particularly rapidly in Asia (Schneider and van der A., 2012) and CO concentrations decreased, most significantly in the Northern Hemisphere (Worden et al., 2013; Yoon and Pozzer, 2014). These trends in $NO_X$ and CO may have contributed to a shift within $HO_X$ from $HO_2$ to OH. Further, Naik et al. (2013) defined the tropospheric domain as extending from the surface up to 200 hPa, whereas we diagnose the tropopause height. In effect Naik et al. include part of the extratropical lower stratosphere, where $\tau_{CH4}$ is about a century. Another reason is that our MOM mechanism more efficiently recycles OH than other VOC chemistry schemes applied in global models. This is supported by our calculation of the MCF lifetime of 5.1 years, which compares with 5.7±0.9 years by Naik et al. (2013), based on a range of 4.1 – 8.4 years among the 17 participating models.

We calculate that at the tropopause and the poles $\tau_{CH4}$ is longest, about a century. The mean $\tau_{CH4}$ in the extra-tropics is 13.8 years and in the inner tropics 6.1 years. The mean $\tau_{CH4}$ in the BL is 4.9 and in the FT 9.1 years. The effective range in the mean OH concentration and $\tau_{CH4}$ between the high- and the low-latitude troposphere is about a factor ten, which is close to the OH and $HO_2$ range between the summer and winter at high latitudes. This is much smaller than the low-to-high latitude gradients and the seasonal cycle of primary OH formation, indicative of the important role of secondary formation (Sec. 6). The NH/SH ratio of $\tau_{CH4}$ is 0.77. Similar differences and latitude contrasts are found for the lifetime of tropospheric CO ($\tau_{CO}$) due to reaction with OH. In the tropics $\tau_{CO}$ is on average about 38 days, in the NH extra-tropics 65 days, in the SH extra-tropics 86 days, and the NH/SH ratio of $\tau_{CO}$ is 0.87.

## 6 Radical budget and recycling probability

Fig. 6 presents a summary of global, annual mean $HO_X$ production terms in the troposphere, also listed in Table 1, which gives an overview of sources and sinks. Primary OH formation by reactions R1,2 (*P*, purple), amounts to 84 Tmol/yr, of which about 85% takes place in the FT. We find that gross OH formation (*G*) and $HO_2$ production in the FT also account for about 85% of the tropospheric total. Secondary OH formation (*S*) in the troposphere adds up to 167 Tmol/yr, i.e., 67% of *G*, the latter being 251 Tmol/yr. *S* is subdivided into contributions by the $NO_X$ mechanism (R7, blue), the $O_X$ mechanism (R11

and R12; green and yellow, respectively) and the OH recycling in VOC chemistry, the OVOC mechanism (red). The result that $r > 60\%$ indicates that global OH is buffered, i.e., not sensitive to chemical perturbations. Fig. 6 illustrates that the fractional contributions by the different production terms in the FT equal those in the troposphere as a whole. It is not surprising that the FT is the dominant reservoir in atmospheric oxidation as it contains 6-7 times more mass than the BL, though it shows that OH formation is rather evenly distributed between different environments within the troposphere, in spite of differences in precursors species and pollution levels.

On a global scale, the relative magnitudes of different OH production terms in the BL and FT are similar (Fig. 6), though the OVOC mechanism (red) is somewhat larger, and the $O_X$ mechanism (green and yellow) somewhat smaller than in the FT. The contribution by the $NO_X$ mechanism, i.e., R7 ($NO+HO_2$, blue), is marginally smaller in the BL (30%) than the FT (31%), in spite that large areas in the BL are more directly influenced by anthropogenic $NO_X$ emissions. As explained above, the contribution of $NO_X$ to OH recycling can be locally self-limiting, e.g., in the strongly polluted BL, while some $NO_X$ – partly as reservoir gases like organic nitrates – can escape to the FT where relatively lower concentrations can be effective in OH production. Examples of $NO_X$ reservoir gases in MOM are alkyl nitrates with carbonyls, e.g., nitro-oxyacetone (NOA) and the nitrate of methyl ethyl ketone.

By comparing gross OH formation $G$ between different regions we find that it is about twice as high in the tropics than the extra-tropics, and 16% lower in the SH than the NH. The upper panel of Fig. 7 presents $G$ in ppbv/day (the lower panels $P$ and $S$), with a global annual average in the troposphere of 4.8 ppbv/day. At low latitudes $G$ is much higher over continents than oceans, related to strong OH recycling, while at high latitudes longitudinal gradients are small, also between oceans and continents in the NH (Fig. 7). Since emissions that affect OH largely occur on land, the latter underscores that on a large scale OH is buffered through processes in the FT. Regional maxima of $G$ are found over the Amazon, Central Africa and southeastern Asia, and smaller areas north of the Mexican Gulf in the USA, Central America and Indonesia (Fig. 7). Over the Amazon and Central Africa we find a relatively high $G$ up to the tropopause, related to deep convection and lightning $NO_X$ over regions that are rich in natural VOCs. Within the BL $G$ can vary greatly, e.g., being on average more than 3 times larger in the CBL than in the MBL. Comparing $P$ between different regions we find that it is 37% higher in the tropics compared to the subtropics, while on average it is the same over oceans and continents.

Consequently, average $S$ is also the same over oceans and continents; though below we underscore that the underlying chemical mechanisms can be very different. In the SH extra-tropics $P$ is about 40% lower than in the NH, mostly associated with the lower abundance of tropospheric $O_3$ in the SH. This inter-hemispheric asymmetry is manifest in the middle panels of Fig. 7. Comparison of the middle and lower panels in Fig. 7 shows that spatial gradients of $P$ and $S$ can be rather different, e.g., towards high latitudes with $P$ falling off with solar radiation and water vapor, while $P$ also declines with altitude. In these regions gradients of $S$ are weaker than of $P$. This actually contributes to OH buffering, as the relatively low rate of $P$ is partly compensated by $S$. This mechanism also acts seasonally, i.e., $S$ is relatively more important in winter.

Rohrer et al. (2006, 2014) emphasized the tight linear relationship between tropospheric OH and UV radiation in Germany and China, expressed by measurements of OH and the photo-dissociation frequency of $O_3$ ($J(O^1D)$). While the

relationship with sunlight is also evident from our results, the interpretation is not straightforward because $P$ also depends on $O_3$ and $H_2O$, and $S$ additionally depends on other factors. For example, in the tropics $P$ has a maximum in the lower troposphere and a minimum in the upper troposphere where the UV intensity is higher, related to dependencies of the $J(O^1D)$ quantum yield and $H_2O$ on temperature. Hence the slope of the regression is different. Furthermore, $S$ is not contingent on $J(O^1D)$ and is generally less strongly dependent on solar radiation.

This is illustrated by Fig. 8, indicating that sometimes a tight linear relationship with $J(O^1D)$ is found, e.g., for $P$ in the BL, but that the relationship with $S$ in the BL is less compact, while in the FT $S$ can deviate from linearity at low UV intensity. Based on a global sample size of 1.45 million pairs from our model calculations, we find a high correlation $R^2=0.94$ between $P$ and $J(O^1D)$, and a lower correlation $R^2=0.80$ between $S$ and $J(O^1D)$. While the mean slope for $P$ is 0.99 (intercept close to zero), it is 0.46 for $S$ (intercept about 0.3). Therefore, there is no unique relationship between OH and UV radiation as it depends on the relative importance of $P$, $S$ and the different mechanisms that contribute to $S$.

Fig. 9 illustrates the efficiency at which OH is recycled, i.e., the recycling probability $r=1-P/G$. We find relatively large differences between tropospheric reservoirs, e.g., between the CBL and MBL, and also between the tropics and extra-tropics. When $S$ is smaller than $P$, $r$ is below 50% (yellow). However, if we consider the troposphere as a whole, $S$ exceeds $P$ everywhere due to the predominance of OH recycling in the FT. In the low latitude MBL $r$ is lowest, indicative of a relatively high sensitivity to perturbations such as large-scale variations and trends in $CH_4$ and CO. This is not the case in the continental troposphere where natural VOCs play an important role in OH recycling. Fig. 9 shows that $r$ is relatively larger in the extra-tropics than in the tropics, and largest at high latitudes.

The chemical buffering mechanisms include the dominant though self-limiting effect of $NO_X$ on OH formation in polluted air, the latter through reaction R10, which is an important sink of both $NO_2$ and OH when concentrations are high ($NO_X$ mechanism; blue in Fig. 6). In unpolluted, low-$NO_X$ conditions the OVOC mechanism acts through competition of unsaturated peroxide and carbonyl sinks, e.g., hydroperoxide-aldehyde (HPALD) in isoprene chemistry (red in Fig. 6). When OH is high, HPALD reacts with OH, whereas at low OH photo-dissociation takes the upper hand through the formation of peroxy-acid aldehyde (PACALD), which produces OH. Over land OH is generally buffered by the $NO_X$ and OVOC mechanisms, illustrated by values of $r$ well over 50% (Fig. 9). However, remote from $NO_X$ and VOC sources in the BL over the tropical and subtropical oceans $r$ can be below 40%. In these environments OH recycling depends on the $O_X$ mechanism (green plus yellow in Fig. 6), which has limited efficiency because R11 ($O_3+HO_2$) is a net oxidant sink. Hence the $O_X$ mechanism depends on replenishment of $O_3$ through transport in the FT and subsequent mixing into the BL.

Differences in $S$ between tropospheric reservoirs, e.g., the CBL, MBL, tropics and extra-tropics, are associated with these three principal OH recycling mechanisms, to various degrees related to natural and anthropogenic VOC and $NO_X$ emissions. Fig. 10 illustrates how OH is buffered both on local and global scales. It shows the fractional contributions of the $NO_X$, $O_X$ and OVOC mechanisms to the overall recycling probability $r$, and indicates that the three mechanisms are complementary. The $NO_X$ mechanism dominates in the NH, especially in polluted air at middle latitudes, and most strongly over the continents. In the SH over the continents, in low-$NO_X$ air, the OVOC mechanism dominates. In the marine

environment – except the pollution outflow regions over the Atlantic and Pacific Oceans – the $O_X$ mechanism predominates. Seasonal complementarity of the three mechanisms is most significant at high latitudes, especially in the BL. Whereas in summer the $O_X$ mechanism is most efficient, and to a lesser degree also the $NO_X$ mechanism, in winter the OVOC mechanism maintains OH formation, being least dependent on solar radiation.

5       To estimate the contributions of the three recycling mechanisms ($NO_X$, $O_X$, OVOC) to global OH and $r$, we performed sensitivity simulations, switching them off one-by-one. By excluding OH recycling by $NO_X$, the global mean OH concentration declines from $11.3 \cdot 10^5$ to $2.7 \cdot 10^5$ molecules/cm$^3$, i.e., a reduction by 76%, while $\tau_{CH4}$ increases from 8.5 to 21.6 years, $r$ reduces from 67% to 42%, and the global mean production of OH drops from 4.8 to 2.8 ppbv/day. This result corroborates the great importance of this mechanism, and the sensitivity of global OH to $NO_X$ abundance. The latter is

illustrated by Fig. 11, which shows zonal mean OH concentrations by the reference simulation and by excluding the three OH recycling mechanisms one-by-one. The $NO_X$ mechanism clearly has the largest impact on global OH, i.e., through the partitioning between OH and $HO_2$ and through the formation of $O_3$. Fig. 11 also shows that model calculated OH exhibits near-interhemispheric parity in the FT, while the $NO_X$ mechanism leads to relatively more OH in the NH, primarily in the subtropical boundary layer.

15       The strength of the $O_X$ mechanism comes second in magnitude, as its omission leads to a drop in global OH from $11.3 \cdot 10^5$ to $5.9 \cdot 10^5$ molecules/cm$^3$, i.e., a reduction by 48%, while $\tau_{CH4}$ increases from 8.5 to 15.0 years, $r$ reduces from 67% to 52%, and the global mean production of OH decreases from 4.8 to 3.4 ppbv/day. The overall strength of the OVOC mechanism is relatively weakest of the three. When we switch it off, global OH decreases from $11.3 \cdot 10^5$ to $9.7 \cdot 10^5$ molecules/cm$^3$, i.e., a reduction by 14%, while $\tau_{CH4}$ increases from 8.5 to 9.7 years, $r$ reduces from 67% to 61%, and the

global mean production of OH decreases from 4.8 to 4.2 ppbv/day. Note that in the latter sensitivity simulation we include OH recycling from $HO_2$ that is produced through OVOC chemistry, which would otherwise contribute to the $NO_X$ and $O_X$ mechanisms. The OH formation through $HO_2$, produced in the breakdown of VOC, accounts for about half the OH recycling by the OVOC mechanism.

## 7 Conclusions

The atmospheric oxidation capacity is generally not sensitive to perturbations that may arise from variations or trends in emissions of natural and anthropogenic origin. This is illustrated by global OH calculations with a large number of chemistry-transport models (Naik et al., 2013; Voulgarakis et al., 2013), where differences between models are larger than between pre-industrial, present and future emission scenarios calculated by the same models. This suggests that model physics and chemistry formulations have a greater impact on calculations of global OH than applying different emission

scenarios of source and sink gases. Results from the EMAC atmospheric chemistry – general circulation model illustrate how a combination of tropospheric chemistry and transport mechanisms buffer OH on a range of scales.

The EMAC model includes the recent Mainz Organics Mechanism (MOM) to comprehensively account for VOC chemistry, including higher generation reaction products, leading to a closed atmospheric budget of reactive carbon. The more realistic description of emissions and complex VOC chemistry in MOM compared to previous models substantially increases OH reactivity, bringing it close to measurements (Nölscher et al., 2016). We also find that in the polluted CBL, notably in the subtropical NH during summer, nighttime VOC chemistry, initiated by reaction with $O_3$, can produce OH concentrations in excess of $10^5$ molecules/cm$^3$, which may be relevant for particle nucleation, for example. Nevertheless, nighttime OH does not contribute significantly to the global atmospheric oxidation capacity (e.g., $\tau_{CH4}$ and $\tau_{CO}$).

Global mean OH concentrations in the BL equal those in the FT and thus the troposphere as a whole ($11.3 \cdot 10^5$ molecules/cm$^3$). Tropospheric column average OH concentrations are highest in the tropics, especially over the Amazon, Central Africa and Southeast Asia. Concentrations of $HO_X$ (OH+$HO_2$) are highest in the CBL over the Amazon, Central Africa, Southeast Asia, and some smaller regions over North Australia, the USA north of the Mexican Gulf and near the Persian Gulf. The latter is related to emissions from the petroleum industry in photochemically polluted air.

While measurement campaigns often focus on the BL, the global distribution and variability of OH and $HO_X$ are dominated by the FT. Long-distance transport processes and OH recycling are most efficient in the FT, whereas BL chemistry is more sensitive to local impacts of reactive carbon emissions. Chemical processes during transport in the FT play an important role in global OH buffering through oxidant transport, notably of ozone. The FT connects with the BL through convective mixing by clouds (latent heating) and entrainment by the diurnal evolution of the BL (sensible heating). The latter is more effective in the continental than in the marine environment.

While $HO_X$ concentrations can diverge strongly over the globe, especially in the BL and between seasons, annual averages in the troposphere vary little, e.g., between the tropics and extra-tropics and between hemispheres. Tropospheric OH is buffered through complementary primary and secondary formation mechanisms throughout seasons, latitudes and altitudes. Globally, secondary OH formation exceeds primary formation – through reactions R1,2 – by about a factor two, leading to an OH recycling probability of 67%, hence global OH is not sensitive to perturbations by natural or anthropogenic emission changes. We find that primary OH formation is tightly related to solar UV radiation intensity, whereas this is much less the case for secondary OH formation. There are three principal pathways of secondary OH formation: the $NO_X$, $O_X$ and OVOC mechanisms.

The $NO_X$ mechanism predominates in anthropogenically influenced environments, causing photochemical smog, and outcompetes the OVOC mechanism concomitant with VOC emissions from vegetation. The $NO_X$ mechanism contributes greatly to global OH and $O_3$. When we switch it off in the model global OH declines by 76% and $\tau_{CH4}$ increases by a factor of 2.5. In regions where $NO_X$ is low the photochemistry of natural VOCs, through the breakdown of OVOC and their reaction products, can govern radical recycling and maintain the atmospheric oxidation capacity associated with undisturbed atmosphere-biosphere interactions. While the OVOC mechanism is important for OH production over forests, excluding it reduces global OH by 14%. In regions where both $NO_X$ and VOC concentrations are low, e.g., in the remote marine

environment and at high latitudes, OH recycling strongly depends on the $O_X$ mechanism. When we switch it off global mean OH drops by 48%.

Recycling mechanisms of OH are important near emission sources of $NO_X$ and VOCs in regions of active photochemistry in the BL, but especially in remote areas and the FT where photochemistry is less active. On large scales ozone is a key buffer of OH. To a lesser degree $NO_X$ reservoir species (e.g., organic nitrates) also play a role. On smaller scales, $H_2O_2$ and OVOCs that release OH upon further reaction and photo-dissociation (e.g., organic peroxides and carbonyls) are important. These short-lived reservoir species govern OH sources and sinks within the column. Ozone, with a lifetime of several weeks in the FT, is central to the atmospheric oxidation capacity through long-distance transport, either from the stratosphere or from photochemically polluted regions, through primary OH formation and OH recycling in natural and anthropogenically influenced atmospheres.

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

**Table 1.** Global, annual mean tropospheric source and sink fluxes of OH (Tmol/yr). Sources and sinks are also specified for the boundary layer and free troposphere.

| Sources/Sinks | BL | FT | Troposphere |
|---|---|---|---|
| $O(^1D)+H_2O$ | 12.5 | 71.5 | 84.0 (33%) |
| $NO+HO_2$ | 10.4 | 66.2 | 76.6 (30%) |
| $O_3+HO_2$ | 3.5 | 30.9 | 34.4 (14%) |
| $H_2O_2+h\nu$ | 2.3 | 22.5 | 24.8 (10%) |
| OVOCs, $ROOH+h\nu$ | 6.6 | 24.8 | 31.4 (13%) |
| *Total OH sources* | 35.3 | 215.9 | 251.2 |
| $OH+HO_Y$[1] | 4.8 | 41.4 | 46.2 (18%) |
| $OH+NO_Y$[2] | 0.8 | 3.3 | 4.1 (1.5%) |
| $OH+CH_4$ | 4.1 | 25.7 | 29.8 (12%) |
| $OH+CO$ | 9.6 | 88.2 | 97.8 (39%) |
| $OH+$other $C_1VOC$[3] | 5.7 | 31.3 | 37.0 (15%) |
| $OH+C_{2+}VOC$[4] | 10.3 | 24.4 | 34.7 (14%) |
| Rest | 0.4 | 1.2 | 1.6 (0.5%) |
| *Total OH sinks* | 35.7 | 215.5 | 251.2 |

[1] $H_2$, $O_3$, $H_2O_2$, radical-radical reactions
[2] $NO$, $NO_2$, $HNO_2$, $HNO_3$, $HNO_4$, ammonia, N-reaction products
[3] VOC with one C-atom (excl. $CH_4$), incl. $CH_3OH$, $C_1$-reaction products
[4] VOC with $\geq 2$ C-atoms, $C_{2+}$-reaction products

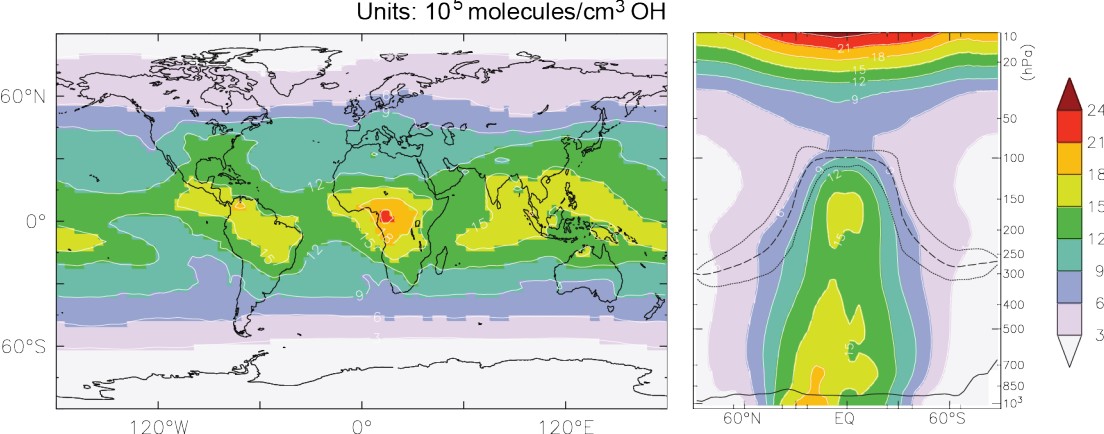

**Figure 1:** Global OH in $10^5$ molecules/cm$^3$. Left: tropospheric, annual mean. Right: zonal, annual mean up to 10 hPa. The lower solid line indicates the average boundary layer height, the upper dashed line the mean tropopause and the solid lines the annual minimum and maximum tropopause height.

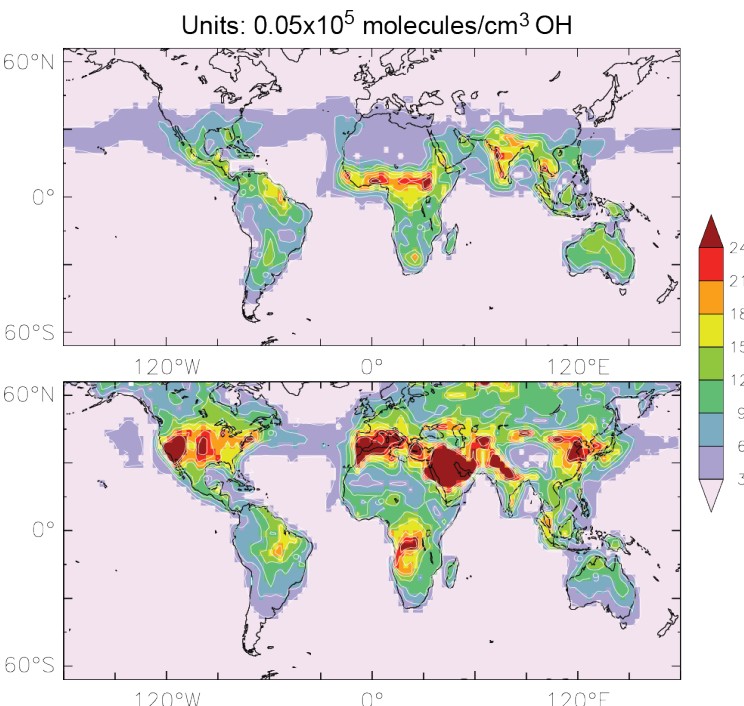

**Figure 2:** Nighttime OH in the boundary layer in January (top) and July (bottom). Color coding is the same as Fig. 1, but concentrations are scaled by a factor 20 (x0.05·$10^5$ molecules/cm$^3$).

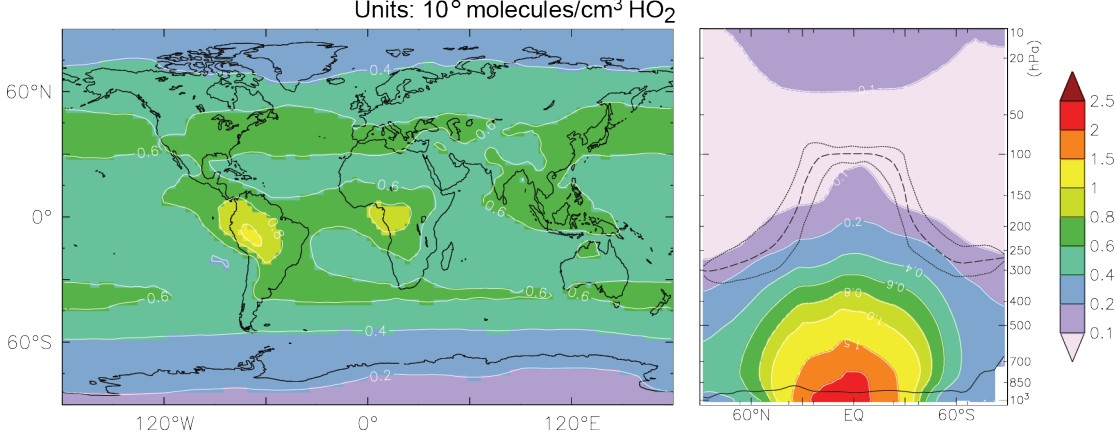

**Figure 3:** As Fig. 1 for $HO_2$ in $10^8$ molecules/cm$^3$ in the troposphere (left) and up to 10 hPa (right).

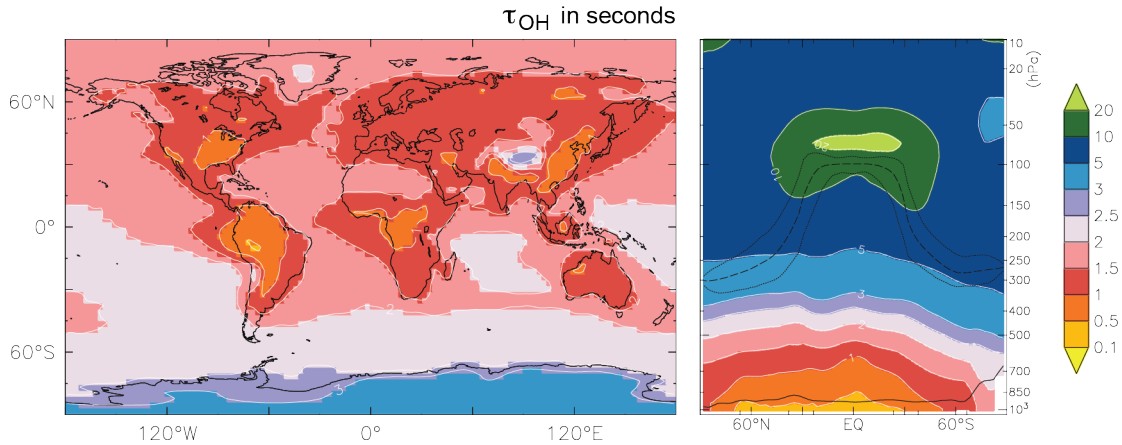

**Figure 4:** As Fig. 1 for the OH lifetime ($\tau_{OH}$, seconds) in the troposphere (left) and up to 10 hPa (right).

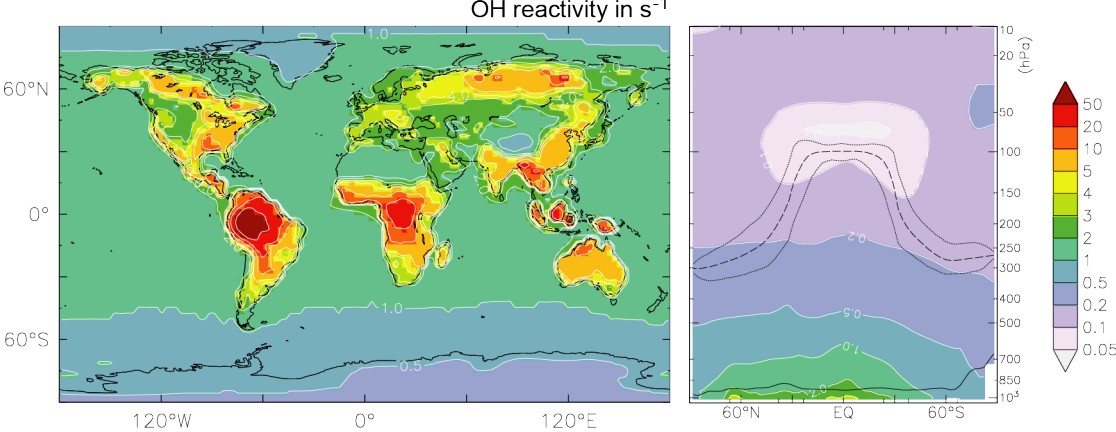

OH reactivity in s⁻¹

**Figure 5:** Annual mean OH reactivity near the Earth's surface in s$^{-1}$.

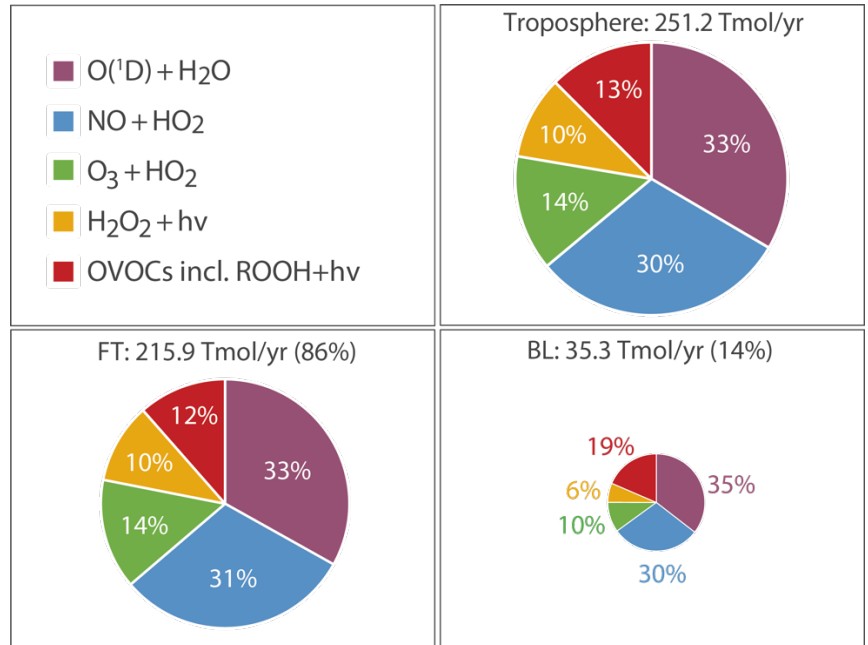

**Figure 6:** Main production terms of OH (Tmol/year) in the troposphere (top right), free troposphere (bottom left) and boundary layer (bottom right). The sizes of the lower two graphs are proportional to the upper right graph, reflecting the percentages of $G$ in parentheses. We distinguish $P$ (purple) from $S$, the latter made up of the $NO_X$ mechanism (blue), the $O_X$ mechanism (yellow and green) and the OVOC mechanism (red).

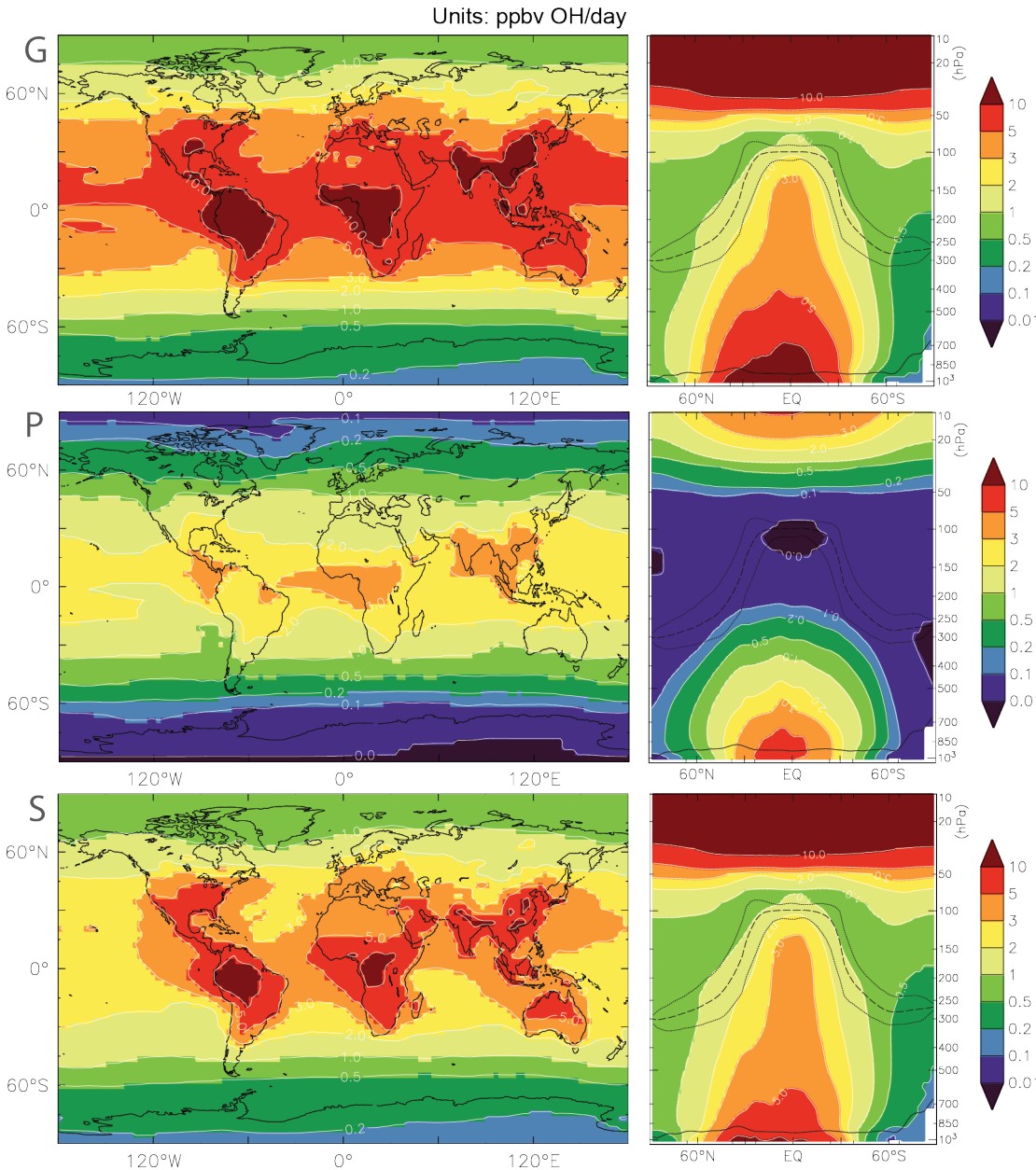

**Figure 7:** Annual mean OH formation in the troposphere (left) and up to 10 hPa (right). The top panels show total (*G*), the middle panels primary (*P*) and the bottom panels secondary (*S*) OH formation (in ppbv/day).

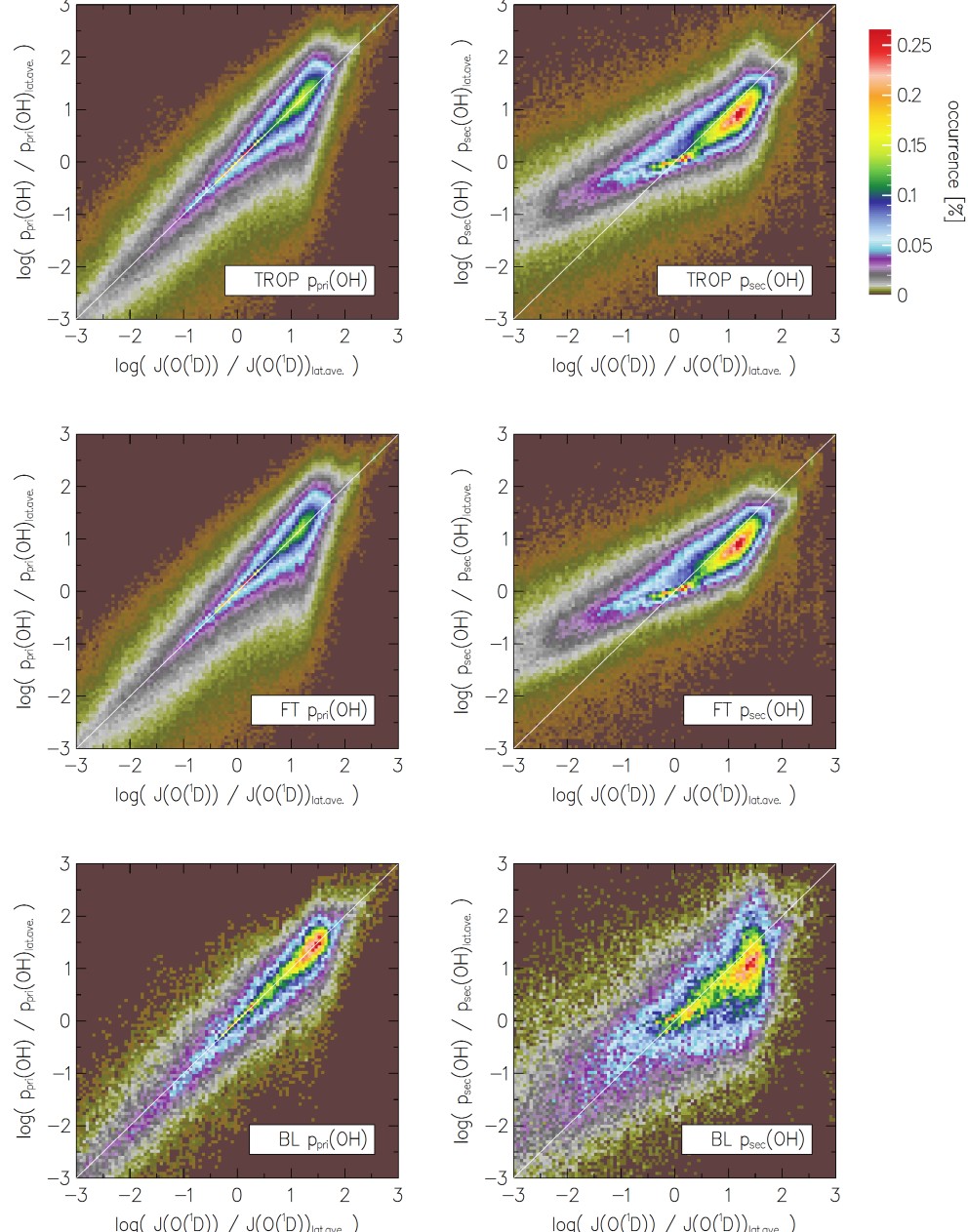

**Figure 8:** Correlation diagrams, showing *P* and *S* on the Y-axes as a function of the photo-dissociation rate of $O_3$ by R1, $J(O^1D)$, on the X-axes. Please notice the log/log scale. *P* is shown in the left panels and *S* in the right panels, in the troposphere (top), FT (middle) and BL (bottom).

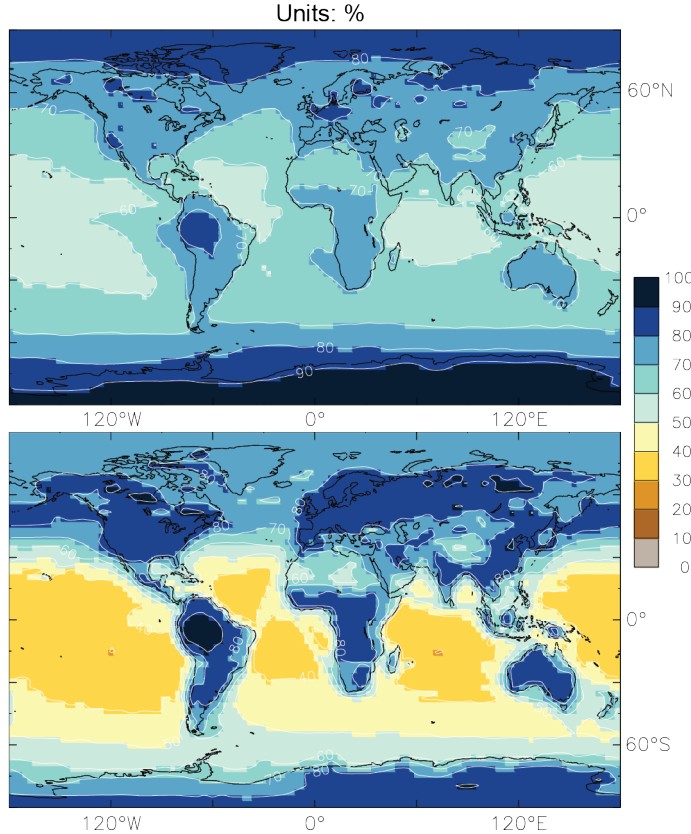

**Figure 9:** Annual mean OH recycling probability ($r$ in %) in the troposphere (top) and the BL (bottom).

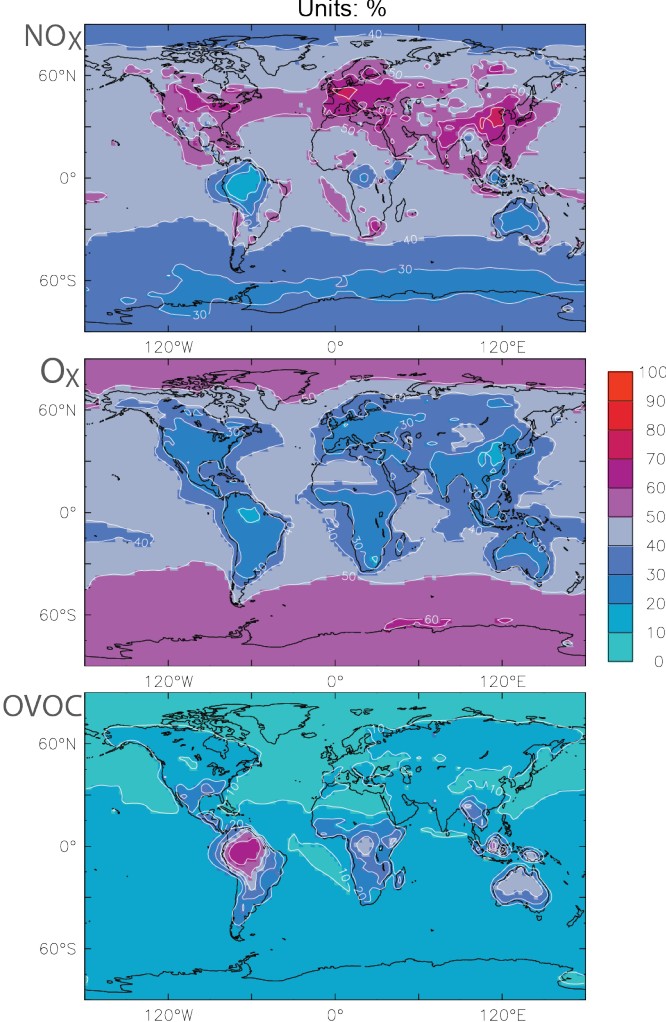

**Figure 10:** Fractional contributions to the OH recycling probability (% of *r*) in the troposphere by the NO$_X$ (top), O$_X$ (middle) and OVOC (bottom) mechanisms (sum of 3 panels is 100%).

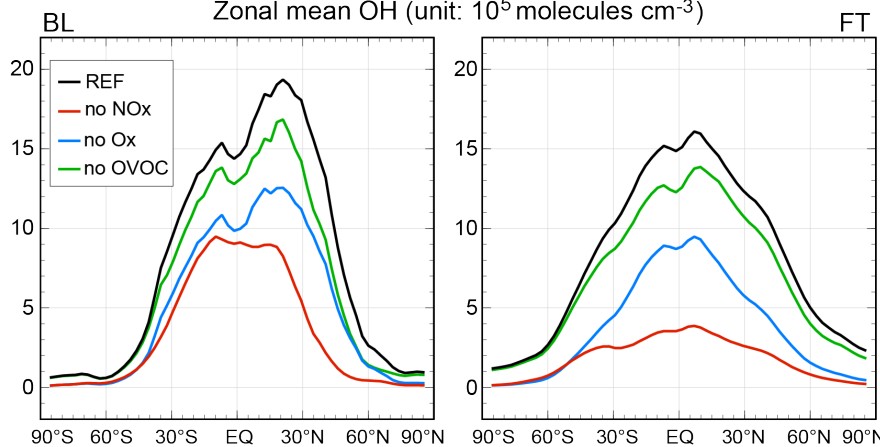

**Figure 11:** Zonal, annual mean OH concentrations calculated in the reference simulation (black) and by successively excluding OH recycling through the $NO_X$, $O_X$ and OVOC mechanisms.