# Peer review of "Global tropospheric hydroxyl distribution, budget and reactivity"

_Atmospheric Chemistry and Physics, 2016_

## Referee Comment (RC1) · Anonymous Referee #1 · 25 Mar 2016

This paper discusses global OH. The paper is well-written and reads smoothly. Illustrations and supplement give a large amount of interesting information. Basically, the paper describes the update of the Mainz-Isoprene-Mechanism (MIM) to MOM and additionally studies the primary formation of OH, the OH recycling (secondary OH formation) as well as HO2. Having read the paper with great pleasure, I looked back and wondered what I learned. This uncovered several fundamental weaknesses of the paper, which are outlined below:

(1) How does MOM change the OH budget?

The paper concentrates on the current model, run for 2013. But I would like to know fundamental things like: By how much did OH increase by including low-NOx recycling? Is the OH abundance still compliant with the CH4 lifetime and/or methyl chloroform

analysis?

(2) Why should we, apart from the OH budget, also analyze the HO2 budget?

The paper analyses the HO2 budget, but without much motivation. Concerning OH, one could argue that this is the "cleansing agent" of our atmosphere. Why HO2, not RO2, or other short-lived species of which the abundances are determined by local chemistry?

(3) The paper discusses concepts like "tele-connections" and "buffering" without proper definitions

Maybe the largest complaint from my side. Earlier work by the authors introduced the "recycling probability", which has at least a proper definition (the probability that OH, once formed, is recycled). However, buffering and tele-connection have not been defined in the paper and hence give freedom to use these terms for everything that is or cannot be fully quantified.

I suggest that the authors at least define and quantify terms like "buffering". One handle could be to actually use perturbations (e.g. the transition from MIM to MOM) to investigate where OH is "buffered". My further remarks below identify parts of the text where the text was specifically vague.

Like stated above, the paper lacks quantitative analysis. In section 4 the HO2 budget is discussed, however, with very little justification (rather vague statements appear: "transport processes influence HOx through longer-lived precursors and reservoir species such as O3 and OVOC"). The authors write: "Our results suggest that HOx is highest over tropical . . .and OH sinks are large". This would call for an analysis in terms of the main photochemical path-ways (i.e. RH, CO shifting the HOx balance towards HO2/RO2, and NO/O3 shifting the balance back through NO/O3 + HO2 OH + NO2, some hints of this analysis in line 26, page 7). Certainly it must be possible to provide a somewhat deeper analysis! Without such an analysis I see very little motivation to show the HO2 budget in such detail. One driver would be the availability of atmospheric observations of HO2, but this is handled by one reference in a short sentence.

Moreover, many statements are made from which the quantitative nature is unclear. Is this speculation, on backed by calculations? Examples: Page 6, lines 26-30: "is partly related to . . ..are a near source aloft" Page 7, lines 28-31: "The efficient atmospheric transport. . .across altitudes"

Vague statements are also given on page 9, lines 7-9: The effective (?) difference in oxidation capacity (is this OH or HOx, how defined?) . . .is a factor of ten, which is close to the extra-tropical seasonal cycle of HOx. This is smaller than . . ., indicative of the important role of secondary (OH) formation. With this statement, the authors seem to suggest that the gradient in OH is linked to the extra-tropical seasonal cycle of HOx, and smaller than the gradient in primary production. It took me a while, however, to decipher this sentence, and once again miss some kind of "interpretation framework" that would deepen general understanding. The main message seems to be that the seasonal cycle in OH is smaller in magnitude than the seasonal cycle in primary OH formation, a statement that is not totally surprising but fro now lacks quantitative explanation and seems poorly connected to concepts like buffering.

Other remarks:

Page 4: Results have been evaluated (page 4, line 22): references are of 2010, after which substantial updates took place in isoprene chemistry. So that must have been different results? Please be clear about validation. The CO and O3 comparisons in the supplement do not look very convincing.

Page 2, line 23: P,S, and G have unit ( moles / year ), please provide.

Page 2, line 32: "observation-based studies". I think it is good to mention specifically that this is based on methyl chloroform, because this suggests "OH observation-based

studies".

Page 4, line 11: "in future". This suggests that this study is based on the complete mechanism. Please say so directly, and mention that this is computationally heavy, and prevents multi-year simulations (i.e. restricting the current study to one year at T42/L31 resolution).

Page 4, line 27: natural VOC emissions are 747-789 TgC/year, but we are discussion 2013 results only. So, it would be correct to give only the 2013 value here.

Page 5, line 11: "may be relevant". I would prefer "are relevant".

Page 5, line 18: "mean tropospheric OH": unclear how troposphere is defined (from the supplement is is clear that a dynamical tropopause is calculated).

Page 6, line 1: "being the main reason". I wonder if this is true. OH in the extra-tropics is much lower than in the tropics. I cannot assess the NH/SH in the tropics, but the influence of "ITCZ"-weighting suggests a leading role of tropical OH.

Page 6, top paragraph. This is now rather confusing. The numbers quoted seem to refer to volume-weighted OH, which puts unrealistic weight on the stratosphere, leading to OH-parity in the integrated atmosphere. This is likely due to parity in the sinks (CO, CH4, etc.) which show much less NH/SH differences in the stratosphere. But I would argue that the mass (or CH4/MCF weighted) OH is the quantity to be analyzed here (see paper Lawrence et al., 2001). Table S15 clearly shows the impact of the selected weighting procedure.

Page 8, line 30-31 and further: Here the authors suggest that the slow rate of SO2 + OH "serves a purpose in the Earth system" (??). This is rather vague again. What I get from it is that, if the reaction rate would be faster at low temperatures, tropical volcanic eruptions would deplete OH completely around the tropopause. This, in turn, would be a threat to the ozone layer, because O3 destroying halocarbons (natural, anthropogenic?) would freely pass the tropopause. If the authors want to suggests

that the ozone layer would not have been formed with a faster SO2 + OH reaction rate in cold conditions in a volcanically-active early Earth, they could simply quantify the impact in their model. Without further elaboration, this side note is clearly out of scope here.

Page 9, lines 31-31: G is the same over the continent as over the oceans, and this would show that OH is buffered through processes in the FT. Figure 7 misses units (mol/m2/year?). This statement implies that the contrast in G of MBL and CBL is large, but these plots are not provided in the supplement. The text mentions that G is on average 3x larger in the CBL than in the MBL, and likely the S-term in G dominates, because the contrast in P is small. Nevertheless, the authors write on page 10, line 7, that S is similar over oceans and continents (in contrast to what figure 7 shows).

Page 10, line 13. This actually contributes to OH buffering. Since there is no actual definition of what "buffering" is, this remains a vague statement. A definition on internet says: "Something that lessens or absorbs the shock of an impact." In the context of OH being driven by primary and secondary formation pathways, I do not see how the "impact" is defined. Only if anything reducing primary production (O3, radiation, water vapor) would result in enhanced recycling of OH, I would see a buffer. A proper definition and analysis of the "buffer" concept would greatly enhance the readability of the paper.

Page 10, line 24. Is the R2 defined for the log-scale or the linear scale? This suggests linear.

Page 10, line 29. Here a new concept is introduced, named a recycling efficiency, defined as (S-P)/G. Together with G = S + P and r = 1-P/G, it seems to me that this representation in abundant, and further complicates the discussion. The information if the left and right panels of figure 9 is therefore similar. I would suggest to avoid further definitions.

Page 11, line 4: "The chemical buffering mechanisms include the dominant though

self-limiting effect of NOx on OH formation in polluted air." Closer to a definition, but now introducing "chemical" and presumably also "transport-related" buffers. It has been shown that at high NOX levels, the chemical system can enter a "run-away" regime (so definitely not buffered), in which fresh NO consumes O3 and the formed NO2 reacts with OH. Given the large grid-cells in EMAC, this effect will likely not occur, but bringing this mechanism as a chemical buffering mechanism seems incorrect to me. Again, the paper would profit from a clear definition and interpretation framework addressing "buffering".

Page 11, discussion figure 10. Now it seems that "r" is defined as "buffer", because figure 10 illustrates how OH is buffered on the local and global scales. Further the authors write: "the complementarity of the three mechanisms in remarkable". Is this not the case by definition as "r" is being decomposed?

Page 12, line 15: "Physical-chemical tele-connections". How are these defined? I am not a fan of the word "teleconnection" because it refers to something that is only vaguely understood (we see a correlation at large distances, but we do not really understand precisely why this correlation is present). In this case atmospheric transport mixes long-lived gases through the atmosphere (O3, CO, CH4, PAN,...) thereby influencing remote regions with "signals" of photochemistry that occurred e.g. over regions with high natural or anthropogenic emissions (e.g. last sentence of the manuscript). This is well-understood, operates on short and long distances, and is therefore definitely not a tele-connection.

---

## Referee Comment (RC2) · Anonymous Referee #2 · 29 Mar 2016

**Review of "Global tropospheric hydroxyl distribution, budget and reactivity", by J. Lelieveld et al.**

**General comments**

This interesting and well written paper describes sources and recycling of OH and $HO_2$ in the EMAC model. In a few places, I found the text could benefit from clarification, and I think some additions/revisions are needed. A full budget of OH is not presented, and the three 'OH recycling' mechanisms are not well defined. I also found it a bit surprising that no comparison with measured values was included. The modelled methane lifetime seems quite short compared to the accepted value – this is a common feature of models, but it goes unmentioned (and I think the comparison with the real world gets worse as more recycling mechanisms are added). Direct comparison to $OH/HO_2$ measurements is also not evident – we are directed to other papers and the model web-site (p4 l22-23), but these papers seem (from their titles) to evaluate other aspects of the model, not HOx, and I couldn't find any mention of the EMAC model version on the web-site. Some statements in the text appear unsubstantiated or a bit over-blown (see specific comments below). Having said all that, I think if these comments and those listed below are adequately addressed, this paper will make a very useful addition to the literature and should be accepted for publication.

**Specific comments**

P1 l26 Spivakovsky

P2 l4 combine, not recombine

P2 l7 This sentence implies RH features in the previous equation(s), but it doesn't?

P2 l23 I find the definition of 'r' (= 1 - P/G) a bit obscure. Isn't r = S/G clearer?

P3 Paragraph from l29 onwards. This paragraph is perhaps not immediately comprehensible to most readers. I suggest you try and make it a bit less technical/more accessible.

P4 l12 …notably FOR carbon…

P4 l30 …in SOME previous… (Not all previous atmospheric chemistry-transport models have had to artificially reduce natural VOC emissions.)

P5 l11 may be -> are

P7 l13 Plane -> Plain

P7 l25 "…tropospheric production of $HO_2$ – and thus HOx…" I don't think this follows. Isn't most $HO_2$ production associated with OH destruction (i.e. HOx recycling, e.g., R3 and R4)? Production of HOx is thus only primary production of either OH or $HO_2$ (i.e. from R1/R2 and HCHO photolysis), whereas production of $HO_2$ is dominated by conversion of OH to $HO_2$. Thus production of HOx and production of $HO_2$ are quite different.

P8 l11 Dividing better than 'relating'.

P8 l23 I wondered what "strongly underestimated" meant here (it is rather non-specific). I would say normally something that is strongly underestimated is 50% or less of its correct value. From your budget in Figure 6, if the VOC reactions were simplified/not included, I don't think OH reactivity would be underestimated by as much as 50%. So I think you are being over-dramatic and non-quantitative, which is unhelpful.

P8 l29 '…indicates that air masses that traverse the TTL into the stratosphere have been largely cleansed from compounds that react with OH'. This seems like overstatement – aren't CO and $CH_4$ the two main compounds that react with OH in the troposphere? I don't think TTL air is 'largely cleansed' of these two gases?

P8 l30 onwards. The 'side note' about $SO_2$, OH, halocarbons and stratospheric $O_3$ seems a bit odd.

P9 l3 onwards. The discussion of the methane lifetime (to oxidation by OH) of 8.5 years in the model should also be compared to observational estimates (e.g., Prather et al., 2012: 11.2 ± 1.3 yr). Pretty much all models, and EMAC with MOM seems not to be an exception, apparently underestimate the methane lifetime. Do we have any idea why this is? It seems that we need less OH in model's atmospheres, but by adding new OH sources from recycling this discrepancy gets worse. Doesn't this suggest that models are missing something fundamental about OH?

P9 l31 '…over the oceans G is the same as over the continents.' G is defined earlier (p2 l23) as 'gross OH formation'. I am unclear whether you mean G over the oceans as a whole compared to G over the continents as a whole, or if you mean per unit area. Obviously this makes a big difference.

P10 l7 '…S is also the same over the oceans and continents…' Same query as previous.

P10 l11 '…P declines steeply with solar radiation and water vapor.' Figure 7 show that P declines steeply with increasing altitude (ignoring the stratosphere) and latitude. Water vapor declines with increasing latitude and altitude (so that's OK). Solar radiation declines with increasing latitude, but increases with increasing altitude. So the relationship of P with solar radiation seems more complex than stated.

P10 l30 'is subordinate to' -> is less than?

P11 l1-2 Isn't 'r' larger in the extra-tropics mainly just because P is small?

P11 l2 The last sentence is true for the MBL but not the CBL, so it is incorrect for the BL as a whole.

P11 l4-13 Some clarification of what is exactly meant by the NOx, $O_3$ and OVOC 'mechanisms' of OH recycling is needed. The earlier reaction equations and discussion is very good and useful, but I am not completely clear on which reactions make up each mechanism.

P11 l13 Do you mean from the FT to the BL (rather than 'transport in the FT')?

P11 l17 'The complementarity of the three mechanisms is remarkable.' Is it? Don't they have to add up to 100% by definition? Figure 10 is certainly interesting, but I am not sure it is 'remarkable'. As suggested earlier, clearer definitions of the three mechanisms would help the discussion.

P12 l4 Have you demonstrated in this paper that including MOM 'increases OH reactivity'? I can believe this is the case, but I don't think you present evidence of what the OH reactivity was in the model before you included MOM.

P12 l13 I note your reference to 'measurement campaigns'. There is no comparison with observations in this paper, which seems like an oversight. Can you demonstrate that modelled OH is improved and compares well to reality?

P18 Table 1. The caption doesn't adequately describe the table – which contains fluxes for HOx primary production ($O^1D+H_2O$), recycling ($NO+HO_2$, $O_3+HO_2$, photolysis reactions) and loss ($H_2O_2$ deposition). This table could be more comprehensive, and describe the full OH and $HO_2$ budgets, i.e. include all the primary sources, OH to $HO_2$ inter-conversions, and sinks (e.g., Derwent, 1996). The

sources and sinks should balance (this is not obvious from the current table). If this were done, it could also clarify the definitions of the three mechanisms, as suggested earlier.

P19 Figure 1 (and all zonal mean plots). It looks like surface pressures go up to 1000 hPa everywhere, but that can't be the case over Antarctica (etc.). Is the vertical scale really pressure?

P21 Figure 6. Related to my comments on Table 1 – I note the caption says 'Main' production terms of OH… Wouldn't a figure that shows *all* the OH sources be more useful?

P24 Figure 9. The text defines recycling efficiency as (S-P)/G. This allows it to take on negative values (where P>S). I find this a bit confusing, as efficiency normally refers to a number between 0-100%. The recycling probability (S/G) does just go from 0-100%. I'm not sure you need both quantities?

P25 Figure 10. Again, clear definitions of the three mechanisms are needed.

**References**

Derwent, R. G. (1996) The Influence of Human Activities on the Distribution of Hydroxyl Radicals in the Troposphere, Phil. Trans. R. Soc. Lond. A, 354, 501-531; DOI: 10.1098/rsta.1996.0018

Prather, M. J., C. D. Holmes, and J. Hsu (2012), Reactive greenhouse gas scenarios: Systematic exploration of uncertainties and the role of atmospheric chemistry, Geophys. Res. Lett., 39, L09803, doi:10.1029/2012GL051440

---

## Referee Comment (RC3) · Anonymous Referee #3 · 29 Mar 2016

This paper discusses the global OH atmospheric chemistry and analyzes the levels and chemical properties and the recycling of OH and $HO_2$ using the global modelling system EMAC. The paper reads smoothly at the most part, providing a lot of information on the chemistry of OH, and an abundance of results both in the main manuscript and the supplementary material.

General Comments

The abstract should be more precise on what the main findings of this work are. The predecessor model as well as the previously assumed amounts of secondary sources should be specifically mentioned here.

[Figure]

The terms "buffering" and "buffered" are used throughout the manuscript without proper definition given. Even after reading the entire manuscript it remains uncertain what the OH buffer actually is.

The entire manuscript is based on the calculations made using an unpublished chemical mechanism (MOM) which is an update of a previous mechanism (MIM) using as reference a manuscript that is in preparation. Even though the full mechanism is included in the Supplementary material of the manuscript, a comparison of the model results using the updated mechanism to results of the previous mechanism and a more detailed comparison to measurements is needed. Also a better more complete budget analysis as well as a comparison and highlight of the differences between the two versions is clearly missing, especially since the authors give relative results such as "higher", or "compared to predecessor models".

The model description is rather short and feels incomplete. The EMAC modeling system is a complex system with a variety of options. The specific sub models used as well as the input used for the present study (i.e. emissions) should be clearly mentioned in the manuscript even if there is a small analysis in the supplementary material. The choice of RCP8.5 that suggests no further emission control also seems strange as it is often used to simulate the worst-case scenario. Even if in the year of interest (2013) the differences from the other scenarios are small, it still is an interesting choice and one that normally should be justified.

Finally I would suggest that a label is added by the colorbar of all figures, indicating the depicted property/substance and the units. This would make the interpretation of the figures quite easier.

Specific Comments

P1, L10: ...may be significant... → change to something more precise, or explain the reason that they might not be significant.

In all the reactions: Add the radical sign (dot) where necessary.

P3, L25: R6 does not directly produce OH, hence a more clear explanation of how OH is produced is needed.

P3, L26: While in polluted air peroxy.... → While in polluted air, peroxy...

P4, L15: By interconnected, do you mean coupled? If yes, the more used (and easier to understand) term should be used. If not, please give a definition of what an interconnected sub model is.

P4, L27/28: Since only one year of results is presented (2013), why is a range of emitted quantities provided?

P5, L24/25 and elsewhere in the manuscript: Add the $10^5$ term to the first number of all ranges.

P5, L27: Give the numbers calculated by Patra et al., since the discussion is based on them.

P7, L4: Reaction R1 of the manuscript should be referenced here.

P7, L7/8 and figure 2 caption: scaled **down** by a factor of 20.

P7, L28/29: $O_3$ from the stratosphere and $O_3$ from photochemically... → $O_3$ from both the stratosphere and photochemically...

P12, L15: The Physical-chemical tele-connections is here used without prior definition. Please give a clear definition.

Figure 5: Add the OH reactivity zonal means (latitudinal) since the height distribution

is mentioned during the discussion in section 5.

Figure 6: Enlarge the third panel of the figure since it is quite difficult to read the numbers in it. Also review the percentages given here since the numbers (as they are provided now) do not add up: e.g. for the OVOCs (red) the FT is 12% and the BL 19%. Multiplied by the 86% and 14% ratios respectively it gives a total of 13% (12.98) in the troposphere, where you present 12%. Maybe give the numbers with at least one decimal point so that the math comes out correct.
* * *

---

## Referee Comment (RC4) · Anonymous Referee #2 · 8 Apr 2016

With respect my earlier comment:

P9 l3 onwards. The discussion of the methane lifetime (to oxidation by OH) of 8.5 years in the model should also be compared to observational estimates (e.g., Prather et al., 2012: 11.2 1.3 yr). Pretty much all models, and EMAC with MOM seems not to be an exception, apparently underestimate the methane lifetime. Do we have any idea why this is? It seems that we need less OH in model's atmospheres, but by adding new OH sources from recycling this discrepancy gets worse. Doesn't this suggest that models are missing something fundamental about OH?

And your reply:

Reply: Prather et al. (2012) derive a CH4 lifetime of 9.10.9 yr [9.1+/-0.1 yr], and indicate this is 5% higher than the multi-model mean, as presented in the IPCC (2007) AR4

assessment, being 8.71.3 yr [8.7 +/-1.3 yr]. Our estimate is somewhat less (8.5 yr) but still consistent.

—

The Prather et al. lifetime of 9.1 +/- 0.1 yr is the total methane lifetime, and includes sinks of methane in the stratosphere, reaction with Cl, and deposition to soils, as well as (the main sink) reaction with OH. Unless I am mistaken, the methane lifetime you quote in the paper is only with respect to reaction with OH. Prather et al. give a breakdown of these components of the total methane lifetime. The methane lifetime with respect to oxidation by OH is 11.2 +/- 1.3 yr (details in the "supporting information", file: grl29135-sup-0006-ts02.xls).

So I believe there to be a discrepancy between the EMAC methane lifetime (to OH) of 8.5 yr and the observationally derived value of 11.2 +/- 1.3 yr.

Feel free to set me straight if you think this is the wrong comparison to make.

Reference:

Prather, M. J., C. D. Holmes, and J. Hsu (2012), Reactive greenhouse gas scenarios: Systematic exploration of uncertainties and the role of atmospheric chemistry, Geophys. Res. Lett., 39, L09803, doi:10.1029/2012GL051440

———————————————————

---

## Author Comment (AC1) · 8 Apr 2016

This paper discusses global OH. The paper is well-written and reads smoothly. Illustrations and supplement give a large amount of interesting information. Basically, the paper describes the update of the Mainz-Isoprene-Mechanism (MIM) to MOM and additionally studies the primary formation of OH, the OH recycling (secondary OH formation) as well as HO2. Having read the paper with great pleasure, I looked back and wondered what I learned. This uncovered several fundamental weaknesses of the paper, which are outlined below:

(1) How does MOM change the OH budget? The paper concentrates on the current model, run for 2013. But I would like to know fundamental things like: By how much did OH increase by including low-NOx recycling? Is the OH abundance still compliant

[Figure]

with the CH4 lifetime and/or methyl chloroform analysis?

Reply: In the revised manuscript we will present sensitivity simulations to demonstrate the influence of the three main OH recycling mechanisms. This will show their influence on OH, including the recycling under low-NOx conditions. It will be interesting to see the difference of OH recycling under high-NOx and low-NOx conditions. In Sect. 5 we discuss the CH4 lifetime in view of results by other models, as presented by Naik et al. (2013). In the revised manuscript we will extend this discussion, including the lifetime of methyl chloroform.

(2) Why should we, apart from the OH budget, also analyze the HO2 budget? The paper analyses the HO2 budget, but without much motivation. Concerning OH, one could argue that this is the "cleansing agent" of our atmosphere. Why HO2, not RO2, or other short-lived species of which the abundances are determined by local chemistry?

Reply: The HO2 budget is included because the main OH conversions in atmospheric chemistry directly relate to HO2 (e.g., reactions with NOx, CO). HO2 is critically important for OH recycling, which is a central topic of our paper. Field and laboratory measurements often address both OH and HO2, e.g., by reporting ratios. By providing global concentration distributions and budgets of both molecules in our paper, we may help guide and interpret radical measurements, while the resulting data in turn provide important constraints for HOx and OH studies. In the revised manuscript we will more explicitly motivate why HO2 has been included.

(3) The paper discusses concepts like "tele-connections" and "buffering" without proper definitions Maybe the largest complaint from my side. Earlier work by the authors introduced the "recycling probability", which has at least a proper definition (the probability that OH, once formed, is recycled). However, buffering and tele-connection have not been defined in the paper and hence give freedom to use these terms for everything that is or cannot be fully quantified. I suggest that the authors at least define and quantify terms like "buffering". One handle could be to actually use perturbations (e.g. the

transition from MIM to MOM) to investigate where OH is "buffered". My further remarks below identify parts of the text where the text was specifically vague.

Reply: We agree that these concepts could be defined more clearly. The concept "tele-connection" is not centrally important in the present context and will be removed from the revised manuscript. However, the concept "buffering" is important and will be discussed more extensively in the revision. This discussion will make use of the model perturbation experiments presented by Lelieveld et al. (2002), i.e., applying NOx and CH4 perturbations and computing the impact on OH. This showed that at an OH recycling probability of 60% or higher, these perturbations have negligible influence on OH. We argue that this can be considered as a buffered chemical system. In the revised manuscript we will also present sensitivity calculations in which the three main OH recycling mechanisms will be switched off one-by-one, which will quantify their role in OH recycling and buffering.

Like stated above, the paper lacks quantitative analysis. In section 4 the HO2 budget is discussed, however, with very little justification (rather vague statements appear: "transport processes influence HOx through longer-lived precursors and reservoir species such as O3 and OVOC"). The authors write: "Our results suggest that HOx is highest over tropical. . .and OH sinks are large". This would call for an analysis in terms of the main photochemical path-ways (i.e. RH, CO shifting the HOx balance towards HO2/RO2, and NO/O3 shifting the balance back through NO/O3 + HO2 OH + NO2, some hints of this analysis in line 26, page 7). Certainly it must be possible to provide a somewhat deeper analysis! Without such an analysis I see very little motivation to show the HO2 budget in such detail. One driver would be the availability of atmospheric observations of HO2, but this is handled by one reference in a short sentence.

Reply: For the HO2 budget, see above. We agree that it would be interesting to study photochemical pathways, and we will try to deepen the analysis, however, without losing scope and our global view of OH, as indicated in the title. Since OH and HO2 are

often studied together for good reason, it makes sense to present both. For example, the lifetime of HO2 helps evaluate the efficiency of OH recycling, which we will make more explicit in the revised manuscript. We believe that providing the budget and distribution of HO2 helps appreciate that of OH.

Moreover, many statements are made from which the quantitative nature is unclear. Is this speculation, on backed by calculations? Examples: Page 6, lines 26-30: "is partly related to. . .are a near source aloft" Page 7, lines 28-31: "The efficient atmospheric transport. . .across altitudes"

Reply: The interpretation of results does not necessarily always needs to be quantitative. This paper aims at conceptual understanding of the atmospheric oxidation efficiency. Certainly this is not speculation. We will refer to previous work that shows the important role of lightning over Central Africa: Tost, H., P. Jöckel and J. Lelieveld (2007) Lightning and convection parameterizations – uncertainties in global modeling. Atmos. Chem. Phys. 7, 4553-4568. For the role of stratosphere – troposphere exchange of ozone we will refer to Lelieveld, J. and F.J. Dentener (2000) What controls tropospheric ozone? J. Geophys. Res. 105, 3531-3551.

Vague statements are also given on page 9, lines 7-9: The effective (?) difference in oxidation capacity (is this OH or HOx, how defined?). . .is a factor of ten, which is close to the extra-tropical seasonal cycle of HOx. This is smaller than. . ., indicative of the important role of secondary (OH) formation. With this statement, the authors seem to suggest that the gradient in OH is linked to the extra-tropical seasonal cycle of HOx, and smaller than the gradient in primary production. It took me a while, however, to decipher this sentence, and once again miss some kind of "interpretation framework" that would deepen general understanding. The main message seems to be that the seasonal cycle in OH is smaller in magnitude than the seasonal cycle in primary OH formation, a statement that is not totally surprising but for now lacks quantitative explanation and seems poorly connected to concepts like buffering.

Reply: The bottom line is that OH recycling can partly compensate the lack of primary OH production in locations and seasons where actinic radiation or water vapor concentrations are low. This is not at all vague, and fits our interpretation framework. However, we agree that the indicated sentence may be difficult to understand. In the revised manuscript we will express this more clearly and elaborately.

Other remarks: Page 4: Results have been evaluated (page 4, line 22): references are of 2010, after which substantial updates took place in isoprene chemistry. So that must have been different results? Please be clear about validation. The CO and O3 comparisons in the supplement do not look very convincing.

Reply: With the sensitivity simulations indicated above, the role of isoprene chemistry will become clearer. We expect to submit a comprehensive ACP manuscript about the new isoprene chemistry soon (Taraborrelli et al., in preparation), and we will update the references.

Page 2, line 23: P,S, and G have unit ( moles / year ), please provide.

Reply: will be done.

Page 2, line 32: "observation-based studies". I think it is good to mention specifically that this is based on methyl chloroform, because this suggests "OH observation-based studies".

Reply: will be done.

Page 4, line 11: "in future". This suggests that this study is based on the complete mechanism. Please say so directly, and mention that this is computationally heavy, and prevents multi-year simulations (i.e. restricting the current study to one year at T42/L31 resolution).

Reply: will be done.

Page 4, line 27: natural VOC emissions are 747-789 TgC/year, but we are discussion

2013 results only. So, it would be correct to give only the 2013 value here.

Reply: will be done.

Page 5, line 11: "may be relevant". I would prefer "are relevant".

Reply: will be done.

Page 5, line 18: "mean tropospheric OH": unclear how troposphere is defined (from the supplement is is clear that a dynamical tropopause is calculated).

Reply: will be improved.

Page 6, line 1: "being the main reason". I wonder if this is true. OH in the extra-tropics is much lower than in the tropics. I cannot assess the NH/SH in the tropics, but the influence of "ITCZ"-weighting suggests a leading role of tropical OH.

Reply: This is correct, as shown in many publications.

Page 6, top paragraph. This is now rather confusing. The numbers quoted seem to refer to volume-weighted OH, which puts unrealistic weight on the stratosphere, leading to OH-parity in the integrated atmosphere. This is likely due to parity in the sinks (CO, CH4, etc.) which show much less NH/SH differences in the stratosphere. But I would argue that the mass (or CH4/MCF weighted) OH is the quantity to be analyzed here (see paper Lawrence et al., 2001). Table S15 clearly shows the impact of the selected weighting procedure.

Reply: It is common to provide the volume weighted OH. To prevent interpretation issues we also give the other metrics suggested by Lawrence et al. (2001). To prevent unrealistic weight on the stratosphere we present results for the troposphere and the lower stratosphere separately.

Page 8, line 30-31 and further: Here the authors suggest that the slow rate of SO2 + OH "serves a purpose in the Earth system" (??). This is rather vague again. What I get from it is that, if the reaction rate would be faster at low temperatures, tropical

volcanic eruptions would deplete OH completely around the tropopause. This, in turn, would be a threat to the ozone layer, because O3 destroying halocarbons (natural, anthropogenic?) would freely pass the tropopause. If the authors want to suggests that the ozone layer would not have been formed with a faster SO2 + OH reaction rate in cold conditions in a volcanically-active early Earth, they could simply quantify the impact in their model. Without further elaboration, this side note is clearly out of scope here.

Reply: We will remove this paragraph from the revised manuscript.

Page 9, lines 31-31: G is the same over the continent as over the oceans, and this would show that OH is buffered through processes in the FT. Figure 7 misses units (mol/m2/year?). This statement implies that the contrast in G of MBL and CBL is large, but these plots are not provided in the supplement. The text mentions that G is on average 3x larger in the CBL than in the MBL, and likely the S-term in G dominates, because the contrast in P is small. Nevertheless, the authors write on page 10, line 7, that S is similar over oceans and continents (in contrast to what figure 7 shows). Page 10, line 13. This actually contributes to OH buffering. Since there is no actual definition of what "buffering" is, this remains a vague statement. A definition on internet says: "Something that lessens or absorbs the shock of an impact." In the context of OH being driven by primary and secondary formation pathways, I do not see how the "impact" is defined. Only if anything reducing primary production (O3, radiation, water vapor) would result in enhanced recycling of OH, I would see a buffer. A proper definition and analysis of the "buffer" concept would greatly enhance the readability of the paper.

Reply: We will include units, and relate the buffering to a definition that will be given in the revised manuscript, as indicated above. We will revise the text in this context.

Page 10, line 24. Is the R2 defined for the log-scale or the linear scale? This suggests linear.

Reply: Correct, this refers to a linear relationship between variables.

Page 10, line 29. Here a new concept is introduced, named a recycling efficiency, defined as (S-P)/G. Together with G = S + P and r = 1-P/G, it seems to me that this representation in abundant, and further complicates the discussion. The information if the left and right panels of figure 9 is therefore similar. I would suggest to avoid further definitions.

Reply: In the revised manuscript we will remove the recycling efficiency (left panels of Fig. 9).

Page 11, line 4: "The chemical buffering mechanisms include the dominant though self-limiting effect of NOx on OH formation in polluted air." Closer to a definition, but now introducing "chemical" and presumably also "transport-related" buffers. It has been shown that at high NOX levels, the chemical system can enter a "run-away" regime (so definitely not buffered), in which fresh NO consumes O3 and the formed NO2 reacts with OH. Given the large grid-cells in EMAC, this effect will likely not occur, but bringing this mechanism as a chemical buffering mechanism seems incorrect to me. Again, the paper would profit from a clear definition and interpretation framework addressing "buffering".

Reply: Runaway conditions have been shown in box modeling studies, while in the real atmosphere such conditions are not observed as they are quickly diluted by transport and mixing processes. This illustrates the importance of transport. Such conditions would also have reduced OH recycling probability and therefore not qualify as buffered. Please realize that even a buffered system can be perturbed so heavily that the buffering capacity is exceeded. Even if runaway conditions would occur locally, they would be of limited duration and not relevant globally. As indicated above, we will provide a clear definition of buffering in the revised manuscript.

Page 11, discussion figure 10. Now it seems that "r" is defined as "buffer", because figure 10 illustrates how OH is buffered on the local and global scales. Further the

authors write: "the complementarity of the three mechanisms in remarkable". Is this not the case by definition as "r" is being decomposed?

Reply: In the revised manuscript we will remove the word remarkable and use "r" to define buffering, as indicated above.

Page 12, line 15: "Physical-chemical tele-connections". How are these defined? I am not a fan of the word "teleconnection" because it refers to something that is only vaguely understood (we see a correlation at large distances, but we do not really understand precisely why this correlation is present). In this case atmospheric transport mixes long-lived gases through the atmosphere (O3, CO, CH4, PAN,...) thereby influencing remote regions with "signals" of photochemistry that occurred e.g. over regions with high natural or anthropogenic emissions (e.g. last sentence of the manuscript). This is well-understood, operates on short and long distances, and is therefore definitely not a tele-connection.

Reply: We will avoid the word tele-connection in the revised manuscript.

---

## Author Comment (AC2) · 8 Apr 2016

This interesting and well written paper describes sources and recycling of OH and HO2 in the EMAC model. In a few places, I found the text could benefit from clarification, and I think some additions/revisions are needed. A full budget of OH is not presented, and the three 'OH recycling' mechanisms are not well defined. I also found it a bit surprising that no comparison with measured values was included. The modelled methane lifetime seems quite short compared to the accepted value – this is a common feature of models, but it goes unmentioned (and I think the comparison with the real world gets worse as more recycling mechanisms are added). Direct comparison to OH/HO2 measurements is also not evident – we are directed to other papers and the model web-site (p4 l22-23), but these papers seem (from their titles) to evaluate other aspects of the model, not HOx, and I couldn't find any mention of the EMAC model version on the

web-site. Some statements in the text appear unsubstantiated or a bit over-blown (see specific comments below). Having said all that, I think if these comments and those listed below are adequately addressed, this paper will make a very useful addition to the literature and should be accepted for publication.

Reply: We will be glad to provide clarifications and additions in the revised manuscript. In the revised manuscript we will present a more comprehensive OH budget and sensitivity simulations to illustrate the impact of the three main OH recycling mechanisms, which will provide a more robust basis for their definition. Comparing with measured OH and HO2 will be difficult, while an evaluation based on the lifetimes of methyl chloroform and methane will be helpful. In addition, we will include a discussion about measured and modeled OH over boreal and tropical forests, being most relevant in the present context. In the revised manuscript these issues will be discussed more elaborately in relation to previous work. We expect to submit a comprehensive ACP manuscript about the new biogenic VOC chemistry soon (Taraborrelli et al., in preparation), and one on the new anthropogenic aromatics chemistry is under review (Cabrera-Perez, D., Taraborrelli, D., Sander, R., and Pozzer, A.: Global atmospheric budget of simple monocyclic aromatic compounds, Atmos. Chem. Phys. Discuss., doi:10.5194/acp-2015-996, in review, 2016.). In the revised manuscript we will update the references.

Specific comments P1 l26 Spivakovsky

Reply: will be changed.

P2 l4 combine, not recombine

Reply: will be changed.

P2 l7 This sentence implies RH features in the previous equation(s), but it doesn't?

Reply: will be reformulated.

P2 l23 I find the definition of 'r' (= 1 - P/G) a bit obscure. Isn't r = S/G clearer?

Reply: In our paper of Lelieveld et al (2002) we defined the recycling probability r by solving the differential equation dOH/dt with a Taylor series expansion. We would like to maintain this definition for consistency.

P3 Paragraph from l29 onwards. This paragraph is perhaps not immediately comprehensible to most readers. I suggest you try and make it a bit less technical/more accessible.

Reply: will be done.

P4 l12 ...notably FOR carbon...

Reply: will be changed.

P4 l30 ...in SOME previous... (Not all previous atmospheric chemistry-transport models have had to artificially reduce natural VOC emissions.)

Reply: will be changed.

P5 l11 may be -> are

Reply: will be changed.

P7 l13 Plane -> Plain

Reply: will be changed.

P7 l25 "...tropospheric production of HO2 – and thus HOx..." I don't think this follows. Isn't most HO2 production associated with OH destruction (i.e. HOx recycling, e.g., R3 and R4)? Production of HOx is thus only primary production of either OH or HO2 (i.e. from R1/R2 and HCHO photolysis), whereas production of HO2 is dominated by conversion of OH to HO2. Thus production of HOx and production of HO2 are quite different.

Reply: good point; will be changed.

P8 l11 Dividing better than 'relating'.

Reply: will be changed.

P8 l23 I wondered what "strongly underestimated" meant here (it is rather non-specific). I would say normally something that is strongly underestimated is 50% or less of its correct value. From your budget in Figure 6, if the VOC reactions were simplified/not included, I don't think OH reactivity would be underestimated by as much as 50%. So I think you are being over-dramatic and non-quantitative, which is unhelpful.

Reply: will be changed.

P8 l29 '...indicates that air masses that traverse the TTL into the stratosphere have been largely cleansed from compounds that react with OH'. This seems like over-statement – aren't CO and CH4 the two main compounds that react with OH in the troposphere? I don't think TTL air is 'largely cleansed' of these two gases?

Reply: Good point; will be changed. We were thinking of compounds that carry halogens into the stratosphere. This will be formulated more explicitly in the revised manuscript.

P8 l30 onwards. The 'side note' about SO2, OH, halocarbons and stratospheric O3 seems a bit odd.

Reply: Also in view of the critical remarks by ref#1 we will remove this paragraph.

P9 l3 onwards. The discussion of the methane lifetime (to oxidation by OH) of 8.5 years in the model should also be compared to observational estimates (e.g., Prather et al., 2012: 11.2 $\pm$ 1.3 yr). Pretty much all models, and EMAC with MOM seems not to be an exception, apparently underestimate the methane lifetime. Do we have any idea why this is? It seems that we need less OH in model's atmospheres, but by adding new OH sources from recycling this discrepancy gets worse. Doesn't this suggest that models are missing something fundamental about OH?

Reply: Prather et al. (2012) derive a CH4 lifetime of 9.1 0.9 yr, and indicate this is 5% higher than the multi-model mean, as presented in the IPCC (2007) AR4 assessment,

being 8.71.3 yr. Our estimate is somewhat less (8.5 yr) but still consistent. Prather et al. (2012) relate their CH4 lifetime to that of methyl chloroform, hence also associated with uncertainty. For example, it is not well known how much methyl chloroform is exchanged with the oceans. Calibrating global OH to a "standard" remains to be a problem, also in view of OH and methyl chloroform distributions. We will probably have to live with some uncertainty, which we nevertheless hope to reduce further in a collaboration project with Wageningen University (Maarten Krol) and NOAA-ESRL (Steve Montzka). We will discuss this issue in greater detail in the revised manuscript.

P9 l31 '...over the oceans G is the same as over the continents.' G is defined earlier (p2 l23) as 'gross OH formation'. I am unclear whether you mean G over the oceans as a whole compared to G over the continents as a whole, or if you mean per unit area. Obviously this makes a big difference.

Reply: This has been averaged and expresses mean G over the oceans and continents, i.e., per unit area. We will express this more clearly in the revised manuscript.

P10 l7 '...S is also the same over the oceans and continents...' Same query as previous.

Reply: This will also be changed accordingly.

P10 l11 '...P declines steeply with solar radiation and water vapor.' Figure 7 show that P declines steeply with increasing altitude (ignoring the stratosphere) and latitude. Water vapor declines with increasing latitude and altitude (so that's OK). Solar radiation declines with increasing latitude, but increases with increasing altitude. So the relationship of P with solar radiation seems more complex than stated.

Reply: We will formulate more accurately in the revised manuscript.

P10 l30 'is subordinate to' -> is less than?

Reply: correct.

P11 l1-2 Isn't 'r' larger in the extra-tropics mainly just because P is small?

Reply: The OH recycling probability would indeed increase if only primary formation would be smaller. However, one expects that primary and secondary OH formation are related and even proportional. The fact that the fraction of secondary relative to primary formation increases with latitude leads to a larger r in the extra-tropics compared to the tropics. This is also the case with altitude. We will formulate this more unambiguously in the revised manuscript.

P11 l2 The last sentence is true for the MBL but not the CBL, so it is incorrect for the BL as a whole.

Reply: Will be corrected.

P11 l4-13 Some clarification of what is exactly meant by the NOx, O3 and OVOC 'mechanisms' of OH recycling is needed. The earlier reaction equations and discussion is very good and useful, but I am not completely clear on which reactions make up each mechanism.

Reply: In the revised manuscript we will define the three mechanisms more clearly, also by sensitivity simulations to show what impact they have on OH.

P11 l13 Do you mean from the FT to the BL (rather than 'transport in the FT')?

Reply: Actually it is both, but we agree that in this context "from" is more adequate.

P11 l17 'The complementarity of the three mechanisms is remarkable.' Is it? Don't they have to add up to 100% by definition? Figure 10 is certainly interesting, but I am not sure it is 'remarkable'. As suggested earlier, clearer definitions of the three mechanisms would help the discussion.

Reply: We will delete the word remarkable and more clearly define the three mechanisms in the revised manuscript.

P12 l4 Have you demonstrated in this paper that including MOM 'increases OH reactivity'? I can believe this is the case, but I don't think you present evidence of what the OH reactivity was in the model before you included MOM.

Reply: We refer to published work, but in the revised manuscript we will discuss the differences in more detail.

P12 l13 I note your reference to 'measurement campaigns'. There is no comparison with observations in this paper, which seems like an oversight. Can you demonstrate that modelled OH is improved and compares well to reality?

Reply: In the revised manuscript we will briefly discuss modeled and measured OH over forested regions (our group has performed and published radical measurements over the boreal and tropical forests), which is most relevant in view of the new MOM mechanism.

P18 Table 1. The caption doesn't adequately describe the table – which contains fluxes for HOx primary production (O1D+H2O), recycling (NO+HO2, O3+HO2, photolysis reactions) and loss (H2O2 deposition). This table could be more comprehensive, and describe the full OH and HO2 budgets, i.e. include all the primary sources, OH to HO2 inter-conversions, and sinks (e.g., Derwent, 1996). The sources and sinks should balance (this is not obvious from the current table). If this were done, it could also clarify the definitions of the three mechanisms, as suggested earlier.

Reply: In the revised manuscript we will adjust the caption and extend the table, as suggested.

P19 Figure 1 (and all zonal mean plots). It looks like surface pressures go up to 1000 hPa everywhere, but that can't be the case over Antarctica (etc.). Is the vertical scale really pressure?

Reply: This is correct. Hence the lower levels over Antarctica in the zonal plots are white. Since we are presenting zonal averages some areas up to 1000 hPa occur almost everywhere, except Antarctica.

[Figure]

P21 Figure 6. Related to my comments on Table 1 – I note the caption says 'Main' production terms of OH... Wouldn't a figure that shows all the OH sources be more useful?

Reply: Considering the comprehensive chemistry scheme in MOM, the figure would become complicated in the VOC part of the pies. This would add little information. The aim of this figure is to provide an overview of the main production terms, i.e., primary formation and the three main OH recycling mechanisms.

P24 Figure 9. The text defines recycling efficiency as (S-P)/G. This allows it to take on negative values (where P>S). I find this a bit confusing, as efficiency normally refers to a number between 0-100%. The recycling probability (S/G) does just go from 0-100%. I'm not sure you need both quantities?

Reply: In the revised manuscript we will remove the recycling efficiency (left panel of Fig. 9).

P25 Figure 10. Again, clear definitions of the three mechanisms are needed.

Reply: This will be remedied in the revised manuscript.

References Derwent, R. G. (1996) The Influence of Human Activities on the Distribution of Hydroxyl Radicals in the Troposphere, Phil. Trans. R. Soc. Lond. A, 354, 501-531; DOI: 10.1098/rsta.1996.0018 Prather, M. J., C. D. Holmes, and J. Hsu (2012), Reactive greenhouse gas scenarios: Systematic exploration of uncertainties and the role of atmospheric chemistry, Geophys. Res. Lett., 39, L09803, doi:10.1029/2012GL051440

---

## Author Comment (AC3) · 8 Apr 2016

This paper discusses the global OH atmospheric chemistry and analyzes the levels and chemical properties and the recycling of OH and HO2 using the global modelling system EMAC. The paper reads smoothly at the most part, providing a lot of information on the chemistry of OH, and an abundance of results both in the main manuscript and the supplementary material.

General Comments The abstract should be more precise on what the main findings of this work are. The predecessor model as well as the previously assumed amounts of secondary sources should be specifically mentioned here.

Reply: The abstract is perhaps too brief. In the revised manuscript we will include some more information about the main findings, OH recycling and secondary sources.

The terms "buffering" and "buffered" are used throughout the manuscript without proper definition given. Even after reading the entire manuscript it remains uncertain what the OH buffer actually is. The entire manuscript is based on the calculations made using an unpublished chemical mechanism (MOM) which is an update of a previous mechanism (MIM) using as reference a manuscript that is in preparation. Even though the full mechanism is included in the Supplementary material of the manuscript, a comparison of the model results using the updated mechanism to results of the previous mechanism and a more detailed comparison to measurements is needed. Also a better more complete budget analysis as well as a comparison and highlight of the differences between the two versions is clearly missing, especially since the authors give relative results such as "higher", or "compared to predecessor models".

Reply: In the revised manuscript we will remedy this. In the predecessor paper of Lelieveld et al. (2002) we presented sensitivity simulations with NOx and CH4, which showed that at a recycling probability of 60% or higher the OH concentration is insensitive to perturbations (i.e., the perturbations that can realistically be expected on Earth). We will use this to define the buffering concept. We will also include a more elaborate OH budget analysis, including a comparison based on methyl chloroform measurements, a discussion of measured and modeled OH over forested areas, and new sensitivity simulations to show the importance of the three main recycling mechanisms (NOx, Ox and OVOC), which will also illustrate the OH recycling in biogenic VOC chemistry introduced into MOM. In the near future we will submit a separate manuscript (by Taraborrelli et al.) to provide details about MOM. This level of technical detail would not fit the format of the present paper.

The model description is rather short and feels incomplete. The EMAC modeling system is a complex system with a variety of options. The specific sub models used as well as the input used for the present study (i.e. emissions) should be clearly mentioned in the manuscript even if there is a small analysis in the supplementary material. The choice of RCP8.5 that suggests no further emission control also seems strange as it is

often used to simulate the worst-case scenario. Even if in the year of interest (2013) the differences from the other scenarios are small, it still is an interesting choice and one that normally should be justified.

Reply: In the revised manuscript we will provide additional model description. Listing the submodels does not help unless a setup is applied in which prognostic routines deviate from the standard (e.g., Jöckel et al., 2010), which is the case here only with the chemistry routine MECCA and the emissions (both provided in the supplement). Actually, the main issue is the namelist used. This can be a problem with a complex model, and we will need to discuss in our EMAC community how to deal with it. One important aspect is that the model version used in the publication is frozen and publicly available. Page 17 of the supplement provides a rather detailed list of emission fluxes applied in the model. While the RCP8.5 scenario is a business-as-usual scenario for $CO_2$ emissions, it is much more conservative for reactive trace gases and generally conceived as more realistic than the other scenarios. For the year of interest (2013) these scenario aspects are hardly relevant.

Finally I would suggest that a label is added by the colorbar of all figures, indicating the depicted property/substance and the units. This would make the interpretation of the figures quite easier.

Reply: In the revised manuscript we will add the labels to the figures.

Specific Comments P1, L10: ...may be significant... ! change to something more precise, or explain thereason that they might not be significant. In all the reactions: Add the radical sign (dot) where necessary.

Reply: In the revised manuscript we will phrase this more precisely. While adding dots to radicals is generally good practice in chemistry, it is unusual in atmospheric chemistry. We will mention it in the text.

P3, L25: R6 does not directly produce OH, hence a more clear explanation of how OH

is produced is needed.

Reply: We will add the complete list of reaction channels for R6, and change

RO2 + HO2 → ROOH + O2 (R6) Into

RO2 + HO2 → ROOH + O2 (R6a)

→ RO + O2 + OH (R6b)

→ ROH + O3 (R6c)

and on p.3 line 25 we will refer to R6b.

P3, L26: While in polluted air peroxy.... ! While in polluted air, peroxy...

Reply: will be done

P4, L15: By interconnected, do you mean coupled? If yes, the more used (and easier to understand) term should be used. If not, please give a definition of what an interconnected sub model is.

Reply: we will replace with coupled.

P4, L27/28: Since only one year of results is presented (2013), why is a range of emitted quantities provided?

Reply: We have performed a 4-year simulation and only show the last year. In the 4-year period the online calculated emissions vary somewhat. We will specify this in the revised manuscript.

P5, L24/25 and elsewhere in the manuscript: Add the 105 term to the first number of all ranges.

Reply: will be done

P5, L27: Give the numbers calculated by Patra et al., since the discussion is based on them.

Reply: will be done

P7, L4: Reaction R1 of the manuscript should be referenced here.

Reply: will be done

P7, L7/8 and figure 2 caption: scaled down by a factor of 20.

Reply: will be mentioned

P7, L28/29: O3 from the stratosphere and O3 from photochemically... ! O3 from both the stratosphere and photochemically...

Reply: will be changed

P12, L15: The Physical-chemical tele-connections is here used without prior definition. Please give a clear definition.

Reply: In the revised manuscript we will refrain from using the work tele-connection.

Figure 5: Add the OH reactivity zonal means (latitudinal) since the height distribution is mentioned during the discussion in section 5.

Reply: This is provided in the supplement. In the revised manuscript we will mention this.

Figure 6: Enlarge the third panel of the figure since it is quite difficult to read the numbers in it. Also review the percentages given here since the numbers (as they are provided now) do not add up: e.g. for the OVOCs (red) the FT is 12% and the BL 19%. Multiplied by the 86% and 14% ratios respectively it gives a total of 13% (12.98) in the troposphere, where you present 12%. Maybe give the numbers with at least one decimal point so that the math comes out correct.

Reply: will be done and corrected where needed.

---

## Author Comment (AC4) · 10 Apr 2016

While we agree with ref#2 that there are discrepancies between estimates of the methane lifetime in the literature, and also that there is a distinct possibility that models overestimate global OH, especially in the northern hemisphere, as indicated by the ACCMIP model inter-comparison, for the time being we should recognize that these estimates constitute the range of uncertainty.

The "observationally" derived lifetime of methane is not available, unfortunately. It could be derived when the global source would be known. However, the current (and past) thinking is that the total methane sink is constrained more accurately, which is used to estimate the global methane source.

Prather et al. (2011) used the best estimate of the atmospheric lifetime of methane.

They also scaled the methyl chloroform (MCF) decay rate to that of methane. This is fine, of course, but there are uncertainties. Firstly, the distributions of both gases are assumed to be the same, which can be discussed, also in view of differences in reaction rates. Secondly, MCF sinks other than through reaction with OH need to be quantified, such as loss in the stratosphere and uptake by the oceans. Actually, the oceans might be a net MCF sink or a source (e.g., Krol and Lelieveld, 2003). Some models account for the stratosphere and others do not. Further, estimates of the total sink of MCF depend on assumptions about MCF emissions, which have been controversially discussed (e.g., Montzka et al., 2011). It is even possible that some small emissions are continuing today.

Also the methane lifetime needs to be corrected for non-OH sinks. The methane loss through reactions with O(1D) and Cl radicals in the stratosphere is relatively well known (about 3% of the methane sink), whereas other sinks, notably uptake by soils, are highly uncertain. Also here it is actually possible that soils are a net methane source rather than a sink.

Overall, we would agree that the methane lifetime needs to be better constrained, and that model results need to be confronted with observations. For this reason we are continuing our efforts to analyze methane and MCF observations. In the revised version of the manuscript we will devote more discussion to this important issue.

References:

Krol, M.C., and J. Lelieveld, Can the variability in tropospheric OH be deduced from measurements of 1,1,1-trichloroethane (methyl chloroform)? J. Geophys. Res., 108, 4125, doi: 10.1029/2002JD002423, 2003.

Montzka, S., M. Krol, E. Dlugokencky, B. Hall, P. Jöckel, and J. Lelieveld, Small inter-annual variability of global atmospheric hydroxyl, Science, 331, 67-69, 2011.

---

## Author Response (AR1)

Ref#1

This paper discusses global OH. The paper is well-written and reads smoothly. Illustrations and supplement give a large amount of interesting information. Basically, the paper describes the update of the Mainz-Isoprene-Mechanism (MIM) to MOM and additionally studies the primary formation of OH, the OH recycling (secondary OH formation) as well as HO2. Having read the paper with great pleasure, I looked back and wondered what I learned. This uncovered several fundamental weaknesses of the paper, which are outlined below:

(1) How does MOM change the OH budget?
The paper concentrates on the current model, run for 2013. But I would like to know fundamental things like: By how much did OH increase by including low-NOx recycling?
Is the OH abundance still compliant with the CH4 lifetime and/or methyl chloroform analysis?

*Reply:*
*We performed sensitivity simulations to determine the role of MOM in the OH budget, as indicated in our reply in the ACPD discussion, to answer the questions of Ref#1. These sensitivity simulations address the three principal OH recycling mechanisms. In Sect. 2 we discuss the new aspects of MOM relative to previous model versions. We added Fig. 11 and the following text to Sect. 6 (plus 3 sentences in the conclusions):*
*"To estimate the contributions of the three recycling mechanisms ($NO_X$, $O_X$, OVOC) to global OH and r, we performed sensitivity simulations, switching them off one-by-one. By excluding OH recycling by $NO_X$, the global mean OH concentration declines from $11.3 \times 10^5$ to $2.7 \times 10^5$ molecules/cm$^3$, i.e., a reduction by 76%, while $\tau_{CH4}$ increases from 8.5 to 21.6 years, r reduces from 67% to 42%, and the global mean production of OH drops from 4.8 to 2.8 ppbv/day. This result corroborates the great importance of this mechanism, and the sensitivity of global OH to $NO_X$ abundance. The latter is also illustrated by Fig. 11, which shows zonal mean OH concentrations by the reference simulation and by excluding the three OH recycling mechanisms one-by-one. The $NO_X$ mechanism clearly has the largest impact on global OH. Fig. 11 also shows that model calculated OH exhibits near-interhemispheric parity in the FT, while the $NO_X$ mechanism leads to relatively more OH in the NH, primarily in the subtropical boundary layer. From Fig. 11 we see that the OH concentrations generated by the different mechanisms do not add up to the reference simulation, because they are complementary and partly compensate the recycling of deactivated mechanisms.*

*The strength of the $O_X$ mechanism comes second in magnitude, as its omission leads to a drop in global OH from $11.3 \times 10^5$ to $5.9 \times 10^5$ molecules/cm$^3$, i.e., a reduction by 48%, while $\tau_{CH4}$ increases from 8.5 to 15.0 years, r reduces from 67% to 52%, and the global mean production of OH decreases from 4.8 to 3.4 ppbv/day. The overall strength of the OVOC mechanism is relatively weakest of the three. When we switch it off, global OH decreases from $11.3 \times 10^5$ to $9.7 \times 10^5$ molecules/cm$^3$, i.e., a reduction by 14%, while $\tau_{CH4}$ increases from 8.5 to 9.7 years, r reduces from 67% to 61%, and the global mean production of OH decreases from 4.8 to 4.2 ppbv/day. Note that in the latter sensitivity simulation we include OH recycling from $HO_2$ that is produced through OVOC chemistry, which would otherwise contribute to the $NO_X$ and $O_X$ mechanisms. The OH formation through $HO_2$, produced in the breakdown of VOC, accounts for about half the OH recycling by the OVOC mechanism."*

*We extended the discussion about the $CH_4$ and methyl chloroform lifetime in Sect. 5 as follows:*
*"Our estimate of the mean lifetime of $CH_4$ due to oxidation by tropospheric OH ($\tau_{CH4}$) is 8.5 years, being within the multi-model calculated $1\sigma$ standard deviation of the mean of $9.7 \pm 1.5$ years presented by Naik et al. (2013), though towards the lower end of the range. Notice that this figure does not include uptake of $CH_4$ by soils and stratospheric loss by OH, $O(^1D)$ and chlorine radicals, which together make up about 10% of the total $CH_4$ sink. The 17 models that participated in the model inter-comparison by Naik et al. (2013) show a range of 7.1 – 14.0 years, while the multi-model mean of 9.7 years was considered to be 5-10% higher than observation-derived estimates.*

*One reason for our $\tau_{CH4}$ estimate being toward the lower end of the range may be that Naik et al. (2013) refer to the year 2000, whereas we applied an emission inventory for the year 2010, i.e., after a*

*period when $NO_X$ concentrations increased particularly rapidly in Asia (Schneider and van der A., 2012) and CO concentrations decreased, most significantly in the Northern Hemisphere (Worden et al., 2013; Yoon and Pozzer, 2014). These trends in $NO_X$ and CO may have contributed to a shift within $HO_X$ from $HO_2$ to OH. Further, Naik et al. (2013) defined the tropospheric domain as extending from the surface up to 200 hPa, whereas we diagnose the tropopause height. In effect Naik et al. include part of the extratropical lower stratosphere, where $\tau_{CH4}$ is about a century. Another reason is that our MOM mechanism more efficiently recycles OH than other VOC chemitry schemes applied in global models. This is supported by our calculation of the MCF lifetime of 5.1 years, which compares with 5.7±0.9 years by Naik et al. (2013), based on a range of 4.1 – 8.4 years among the 17 participating models."*

(2) Why should we, apart from the OH budget, also analyze the HO2 budget?
The paper analyses the HO2 budget, but without much motivation. Concerning OH, one could argue that this is the "cleansing agent" of our atmosphere. Why HO2, not RO2, or other short-lived species of which the abundances are determined by local chemistry?

*Reply: The HO2 budget is included because the main OH conversions in atmospheric chemistry directly relate to HO2 (e.g., reactions with NOx, CO). HO2 is critically important for OH recycling, which is central in our paper. Field and laboratory measurements often address both OH and HO2, e.g., by reporting ratios. By providing global concentration distributions and budgets of both molecules, we may help guide and interpret radical measurements, while the resulting data in turn provide important constraints for HOx and OH studies. To meet the comment of Ref#1, we removed the $HO_2$ budget calculations from Table 1. In the revised manuscript we have added the following text in Sect. 4:*
*"Since conversions between $HO_2$ and OH play a key role in OH recycling, we address the budget of $HO_X$ (OH+$HO_2$), which is dominated by $HO_2$. Field and laboratory measurements often address both OH and $HO_2$."*

(3) The paper discusses concepts like "tele-connections" and "buffering" without proper definitions
Maybe the largest complaint from my side. Earlier work by the authors introduced the "recycling probability", which has at least a proper definition (the probability that OH, once formed, is recycled). However, buffering and tele-connection have not been defined in the paper and hence give freedom to use these terms for everything that is or cannot be fully quantified.
I suggest that the authors at least define and quantify terms like "buffering". One handle could be to actually use perturbations (e.g. the transition from MIM to MOM) to investigate where OH is "buffered". My further remarks below identify parts of the text where the text was specifically vague.

*Reply: We agree that these concepts could be defined more clearly. The concept "tele-connection" is not centrally important in the present context and has been removed from the revised manuscript. However, the concept "buffering" is important and is defined by adding the following text to the introduction:*
*"Lelieveld et al. (2002) performed perturbation simulations of $NO_X$ and $CH_4$ to compute the impact on OH. This showed that at an OH recycling probability of 60% or higher, these perturbations have negligible influence on OH (their Fig. 6). Therefore, at r > 60% the atmospheric chemical system can be considered as buffered."*

Like stated above, the paper lacks quantitative analysis. In section 4 the HO2 budget is discussed, however, with very little justification (rather vague statements appear: "transport processes influence HOx through longer-lived precursors and reservoir species such as O3 and OVOC"). The authors write: "Our results suggest that HOx is highest over tropical…and OH sinks are large". This would call for an analysis in terms of the main photochemical path-ways (i.e. RH, CO shifting the HOx balance towards HO2/RO2, and NO/O3 shifting the balance back through NO/O3 + HO2 OH + NO2, some hints of this analysis in line 26, page 7). Certainly it must be possible to provide a somewhat deeper analysis! Without such an analysis I see very little motivation to show the HO2 budget in such detail. One driver would be the availability of atmospheric observations of HO2, but

this is handled by one reference in a short sentence.

*Reply: For the HO2 budget, see above. We agree that it would be interesting to study photochemical pathways, however, without losing scope and our global view of OH, as indicated in the title. Since OH and HO2 are often studied together for good reason, it makes sense to present both. For example, the lifetime of HO2 helps evaluate the efficiency of OH recycling, which has been made more explicit in the revised manuscript. We believe that providing the budget and distribution of HO2 helps appreciate that of OH.*

*We have deepened the analysis of OH recycling mechanisms by presenting sensitivity simulations of the three main photochemical pathways, also mentioning the contribution of HO2 in OH recycling in VOC chemistry, and by adding and discussing Fig. 11 (text, see above). In the last paragraphs of Sect. 6 we have added quantitative information about the OH recycling mechanisms.*

Moreover, many statements are made from which the quantitative nature is unclear. Is this speculation, on backed by calculations? Examples: Page 6, lines 26-30: "is partly related to…are a near source aloft" Page 7, lines 28-31: "The efficient atmospheric transport…across altitudes"

*Reply: The interpretation of these results does not necessarily needs to be quantitative and builds on prior articles. This paper aims at conceptual understanding of the atmospheric oxidation efficiency. Certainly this is not speculation. In the revision we refer to previous work that shows the important role of lightning over Central Africa: Tost, H., P. Jöckel and J. Lelieveld (2007) Lightning and convection parameterizations – uncertainties in global modeling. Atmos. Chem. Phys. 7, 4553-4568. For the role of stratosphere – troposphere exchange of ozone we refer to Lelieveld, J. and F.J. Dentener (2000) What controls tropospheric ozone? J. Geophys. Res. 105, 3531-3551, and for the marine boundary layer to De Laat, A.T.J. and J. Lelieveld (2000) Diurnal ozone cycle in the marine boundary layer. J. Geophys. Res. 105, 11547-11559.*

Vague statements are also given on page 9, lines 7-9: The effective (?) difference in oxidation capacity (is this OH or HOx, how defined?)…is a factor of ten, which is close to the extra-tropical seasonal cycle of HOx. This is smaller than…, indicative of the important role of secondary (OH) formation. With this statement, the authors seem to suggest that the gradient in OH is linked to the extra-tropical seasonal cycle of HOx, and smaller than the gradient in primary production. It took me a while, however, to decipher this sentence, and once again miss some kind of "interpretation framework" that would deepen general understanding. The main message seems to be that the seasonal cycle in OH is smaller in magnitude than the seasonal cycle in primary OH formation, a statement that is not totally surprising but for now lacks quantitative explanation and seems poorly connected to concepts like buffering.

*Reply: The bottom line is that OH recycling can partly compensate the lack of primary OH production in locations and seasons where actinic radiation or water vapor concentrations are low. For clarity we changed the sentence into:*
*"The effective range in the mean OH concentration and $\tau_{CH4}$ between the high- and the low-latitude troposphere is about a factor ten, which is close to the OH and $HO_2$ range between the summer and winter at high latitudes. This is much smaller than the low-to-high latitude gradients and the seasonal cycle of primary OH formation, indicative of the important role of secondary formation".*

Other remarks:
Page 4: Results have been evaluated (page 4, line 22): references are of 2010, after which substantial updates took place in isoprene chemistry. So that must have been different results? Please be clear about validation. The CO and O3 comparisons in the supplement do not look very convincing.

*Reply: With the three sensitivity simulations indicated above, the role of isoprene and VOC chemistry has become much clearer. We will submit a comprehensive ACP manuscript about the new isoprene chemistry soon (Taraborrelli et al., in preparation), and have updated the references. Part of the new scheme has recently been published by Cabrera-Perez et al. (2016), also using MOM. We have*

*extended the part on validation, e.g., by comparing with OH measurements and OH reactivity over tropical forests and by discussing the calculated MCF lifetime (see above).*

Page 2, line 23: P,S, and G have unit ( moles / year ), please provide.

*Reply: done.*

Page 2, line 32: "observation-based studies". I think it is good to mention specifically that this is based on methyl chloroform, because this suggests "OH observation-based studies".

*Reply: done.*

Page 4, line 11: "in future". This suggests that this study is based on the complete mechanism. Please say so directly, and mention that this is computationally heavy, and prevents multi-year simulations (i.e. restricting the current study to one year at T42/L31 resolution).

*Reply: done.*

Page 4, line 27: natural VOC emissions are 747-789 TgC/year, but we are discussion 2013 results only. So, it would be correct to give only the 2013 value here.

*Reply: done.*

Page 5, line 11: "may be relevant". I would prefer "are relevant".

*Reply: done.*

Page 5, line 18: "mean tropospheric OH": unclear how troposphere is defined (from the supplement is is clear that a dynamical tropopause is calculated).

*Reply: done.*

Page 6, line 1: "being the main reason". I wonder if this is true. OH in the extra-tropics is much lower than in the tropics. I cannot assess the NH/SH in the tropics, but the influence of "ITCZ"-weighting suggests a leading role of tropical OH.

*Reply: This is correct, as shown in previous publications. This is also illustrated by Fig. 1.*

Page 6, top paragraph. This is now rather confusing. The numbers quoted seem to refer to volume-weighted OH, which puts unrealistic weight on the stratosphere, leading to OH-parity in the integrated atmosphere. This is likely due to parity in the sinks (CO, CH4, etc.) which show much less NH/SH differences in the stratosphere. But I would argue that the mass (or CH4/MCF weighted) OH is the quantity to be analyzed here (see paper Lawrence et al., 2001). Table S15 clearly shows the impact of the selected weighting procedure.

*Reply: It is common practice to provide the volume weighted OH. To prevent interpretation issues we also give the other metrics suggested by Lawrence et al. (2001). To prevent unrealistic weight on the stratosphere we concentrate on the troposphere and present results for the lower stratosphere separately.*

Page 8, line 30-31 and further: Here the authors suggest that the slow rate of SO2 + OH "serves a purpose in the Earth system" (??). This is rather vague again. What I get from it is that, if the reaction rate would be faster at low temperatures, tropical volcanic eruptions would deplete OH completely around the tropopause. This, in turn, would be a threat to the ozone layer, because O3 destroying halocarbons (natural, anthropogenic?) would freely pass the tropopause. If the authors want to suggests that the ozone layer would not have been formed with a faster SO2 + OH reaction rate

in cold conditions in a volcanically-active early Earth, they could simply quantify the impact in their model. Without further elaboration, this side note is clearly out of scope here.

*Reply: The paragraph has been removed in the revised manuscript.*

Page 9, lines 31-31: G is the same over the continent as over the oceans, and this would show that OH is buffered through processes in the FT. Figure 7 misses units (mol/m2/year?). This statement implies that the contrast in G of MBL and CBL is large, but these plots are not provided in the supplement. The text mentions that G is on average 3x larger in the CBL than in the MBL, and likely the S-term in G dominates, because the contrast in P is small. Nevertheless, the authors write on page 10, line 7, that S is similar over oceans and continents (in contrast to what figure 7 shows). Page 10, line 13. This actually contributes to OH buffering. Since there is no actual definition of what "buffering" is, this remains a vague statement. A definition on internet says: "Something that lessens or absorbs the shock of an impact." In the context of OH being driven by primary and secondary formation pathways, I do not see how the "impact" is defined. Only if anything reducing primary production (O3, radiation, water vapor) would result in enhanced recycling of OH, I would see a buffer. A proper definition and analysis of the "buffer" concept would greatly enhance the readability of the paper.

*Reply: We have included units (ppbv/day), and relate the buffering to the definition that is given in the introduction of the revised manuscript (r >60%). We agree that the sentence on land-ocean contrasts was confusing, now replaced by:*
*"At low latitudes G is much higher over continents than oceans, related to strong OH recycling, while at high latitudes longitudinal gradients are small, also between oceans and continents in the NH (Fig. 7)."*

Page 10, line 24. Is the R2 defined for the log-scale or the linear scale? This suggests linear.

*Reply: Correct, this refers to a linear relationship between variables.*

Page 10, line 29. Here a new concept is introduced, named a recycling efficiency, defined as (S-P)/G. Together with G = S + P and r = 1-P/G, it seems to me that this representation in abundant, and further complicates the discussion. The information if the left and right panels of figure 9 is therefore similar. I would suggest to avoid further definitions.

*Reply: In the revised manuscript we have removed the recycling efficiency (left panels of Fig. 9).*

Page 11, line 4: "The chemical buffering mechanisms include the dominant though self-limiting effect of NOx on OH formation in polluted air." Closer to a definition, but now introducing "chemical" and presumably also "transport-related" buffers. It has been shown that at high NOX levels, the chemical system can enter a "run-away" regime (so definitely not buffered), in which fresh NO consumes O3 and the formed NO2 reacts with OH. Given the large grid-cells in EMAC, this effect will likely not occur, but bringing this mechanism as a chemical buffering mechanism seems incorrect to me. Again, the paper would profit from a clear definition and interpretation framework addressing "buffering".

*Reply: Runaway conditions have been shown in box modeling studies, while in the real atmosphere such conditions are not observed as they are quickly diluted by transport and mixing processes. This illustrates the importance of transport and mixing. Such conditions would also have reduced OH recycling probability and therefore not qualify as buffered. Please realize that even a buffered system can be perturbed so heavily that the buffering capacity is exceeded. Even if runaway conditions would occur locally, they would be of short duration and not relevant globally. In the revised manuscript we have provided an unambiguous definition of buffering.*

Page 11, discussion figure 10. Now it seems that "r" is defined as "buffer", because figure 10 illustrates how OH is buffered on the local and global scales. Further the authors write: "the complementarity of the three mechanisms in remarkable". Is this not the case by definition as "r" is being decomposed?

*Reply: In the revised manuscript we removed the word remarkable and use "r" to define buffering, as indicated above. The text has been revised to:*
*"Fig. 10 illustrates how OH is buffered both on local and global scales. It shows the fractional contributions of the $NO_X$, $O_X$ and OVOC mechanisms to the overall recycling probability r, and indicates that the three mechanisms are complementary."*

Page 12, line 15: "Physical-chemical tele-connections". How are these defined? I am not a fan of the word "teleconnection" because it refers to something that is only vaguely understood (we see a correlation at large distances, but we do not really understand precisely why this correlation is present). In this case atmospheric transport mixes long-lived gases through the atmosphere (O3, CO, CH4, PAN,…) thereby influencing remote regions with "signals" of photochemistry that occurred e.g. over regions with high natural or anthropogenic emissions (e.g. last sentence of the manuscript). This is well-understood, operates on short and long distances, and is therefore definitely not a tele-connection.

*Reply: We removed reference to tele-connections in the revised manuscript.*

Ref#2

This interesting and well written paper describes sources and recycling of OH and HO2 in the EMAC model. In a few places, I found the text could benefit from clarification, and I think some additions/revisions are needed. A full budget of OH is not presented, and the three 'OH recycling' mechanisms are not well defined. I also found it a bit surprising that no comparison with measured values was included. The modelled methane lifetime seems quite short compared to the accepted value – this is a common feature of models, but it goes unmentioned (and I think the comparison with the real world gets worse as more recycling mechanisms are added). Direct comparison to OH/HO2 measurements is also not evident – we are directed to other papers and the model web-site (p4 l22-23), but these papers seem (from their titles) to evaluate other aspects of the model, not HOx, and I couldn't find any mention of the EMAC model version on the web-site. Some statements in the text appear unsubstantiated or a bit over-blown (see specific comments below). Having said all that, I think if these comments and those listed below are adequately addressed, this paper will make a very useful addition to the literature and should be accepted for publication.

*In the revised manuscript we added a full OH budget, more clearly define the three OH recycling mechanisms, and extended the discussion of OH model/measurement results over tropical forest.*
*As also requested by ref#1, we expanded the discussion on the lifetime of methane, indicating potential discrepancies with other work. Sect. 5 has been amended as follows:*
*"Our estimate of the tropospheric mean lifetime of $CH_4$ due to oxidation by OH ($\tau_{CH4}$) is 8.5 years, being within the multi-model calculated $1\sigma$ standard deviation of the mean of 9.7±1.5 years presented by Naik et al. (2013), though towards the lower end of the range. Notice that this figure does not include uptake of $CH_4$ by soils and stratospheric loss by OH, $O(^1D)$ and chlorine radicals, which together make up about 10% of the total $CH_4$ sink. The 17 models that participated in the model inter-comparison by Naik et al. (2013) show a range of 7.1 – 14.0 years, while the multi-model mean of 9.7 years was considered to be 5-10% higher than observation-derived estimates.*

*One reason for our $\tau_{CH4}$ estimate being toward the lower end of the range may be that Naik et al. (2013) refer to the year 2000, whereas we applied an emission inventory for the year 2010, i.e., after a period when $NO_X$ concentrations increased particularly rapidly in Asia (Schneider and van der A., 2012) and CO concentrations decreased, most significantly in the Northern Hemisphere (Worden et al., 2013; Yoon and Pozzer, 2014). These trends in $NO_X$ and CO may have contributed to a shift within $HO_X$ from $HO_2$ to OH. Further, Naik et al. (2013) defined the tropospheric domain as extending from the surface up to 200 hPa, whereas we diagnose the tropopause height. In effect Naik et al. include part of the extratropical lower stratosphere, where $\tau_{CH4}$ is about a century. Another reason is that our MOM mechanism more efficiently recycles OH than other chemical schemes applied in global models. This is supported by our calculation of the MCF lifetime of 5.1 years, which compares with 5.7±0.9 years by Naik et al. (2013), based on a range of 4.1 – 8.4 years among the 17 participating models."*

*With respect to OH over the forest, we added the following text:*
*"In the BL over tropical forests OH concentrations are comparatively low, about $10\times10^5$ to $20\times10^5$ molecules/$cm^3$, in agreement with OH measurements in South America and Southeast Asia (Kubistin et al., 2010; Pugh et al., 2010; Whalley et al., 2011)."*

*We will submit a comprehensive ACP manuscript about the new biogenic VOC chemistry soon (Taraborrelli et al., in preparation), while one on the new anthropogenic aromatics chemistry has been published by Cabrera-Perez et al. (2016) who also use MOM.*

Specific comments
P1 l26 Spivakovsky

*Reply: changed.*

P2 l4 combine, not recombine

*Reply: changed.*

P2 l7 This sentence implies RH features in the previous equation(s), but it doesn't?

*Reply: has been reformulated.*

P2 l23 I find the definition of 'r' (= 1 - P/G) a bit obscure. Isn't r = S/G clearer?

*Reply: In our paper of Lelieveld et al (2002) we defined the recycling probability r by solving the differential equation dOH/dt with a Taylor series expansion. We would like to maintain this definition for consistency.*

P3 Paragraph from l29 onwards. This paragraph is perhaps not immediately comprehensible to most readers. I suggest you try and make it a bit less technical/more accessible.

*Reply: We changed it into:*
*"An important pathway in isoprene chemistry, basic to the recycling of OH, is isomerization through H-migration within oxygenated reaction products, leading to photo-labile hydroperoxy-aldehydes (HPALD), as reviewed by Vereecken and Francisco (2012)."*

P4 l12 …notably FOR carbon…

*Reply: changed.*

P4 l30 …in SOME previous… (Not all previous atmospheric chemistry-transport models have had to artificially reduce natural VOC emissions.)

*Reply: changed.*

P5 l11 may be -> are

*Reply: changed.*

P7 l13 Plane -> Plain

*Reply: changed.*

P7 l25 "…tropospheric production of HO2 – and thus HOx…" I don't think this follows. Isn't most HO2 production associated with OH destruction (i.e. HOx recycling, e.g., R3 and R4)? Production of HOx is thus only primary production of either OH or HO2 (i.e. from R1/R2 and HCHO photolysis), whereas production of HO2 is dominated by conversion of OH to HO2. Thus production of HOx and production of HO2 are quite different.

*Reply: changed.*

P8 l11 Dividing better than 'relating'.

*Reply: changed.*

P8 l23 I wondered what "strongly underestimated" meant here (it is rather non-specific). I would say normally something that is strongly underestimated is 50% or less of its correct value. From your budget in Figure 6, if the VOC reactions were simplified/not included, I don't think OH reactivity would be underestimated by as much as 50%. So I think you are being over-dramatic and non-quantitative, which is unhelpful.

*Reply: Actually we are not exaggerating, as Mogensen et al (2015) identified "missing OH reactivity" in the boreal forest to be about 65% and Whalley et al. (2011) and Nölscher et al. (2016) up to a*

*factor 10 in the tropical forest. We have added the references to Mogensen et al. and Whalley et al. and the numeric value of a factor of 10 to the revised manuscript.*

P8 l29 '…indicates that air masses that traverse the TTL into the stratosphere have been largely cleansed from compounds that react with OH'. This seems like overstatement – aren't CO and CH4 the two main compounds that react with OH in the troposphere? I don't think TTL air is 'largely cleansed' of these two gases?

*Reply: The formulation has been changed into:*
*"While this is largely related to low temperatures and reduced reaction rates, it also indicates that air masses that traverse the tropical transition layer into the stratosphere are cleansed from reactive compounds that are removed by OH, for example organo-halogen compounds that could damage the ozone layer."*

P8 l30 onwards. The 'side note' about SO2, OH, halocarbons and stratospheric O3 seems a bit odd.

*Reply: Also in view of the critical remarks by Ref#1 this paragraph has been removed.*

P9 l3 onwards. The discussion of the methane lifetime (to oxidation by OH) of 8.5 years in the model should also be compared to observational estimates (e.g., Prather et al., 2012: 11.2 ± 1.3 yr). Pretty much all models, and EMAC with MOM seems not to be an exception, apparently underestimate the methane lifetime. Do we have any idea why this is? It seems that we need less OH in model's atmospheres, but by adding new OH sources from recycling this discrepancy gets worse. Doesn't this suggest that models are missing something fundamental about OH?

*Reply: Prather et al. (2012) derive a $CH_4$ lifetime of 9.1±0.9 yr, and indicate this is 5% higher than the multi-model mean, as presented in the IPCC (2007) AR4 assessment, being 8.7±1.3 yr. Our estimate is somewhat less (8.5 yr) but still consistent. Prather et al. (2012) relate their $CH_4$ lifetime to that of methyl chloroform, hence also associated with uncertainty. For example, it is not well known how much methyl chloroform is exchanged with the oceans. Calibrating global OH to a "standard" remains to be a problem, also in view of OH and methyl chloroform distributions. We will probably have to live with some uncertainty, which we nevertheless hope to reduce further in a collaboration project with Wageningen University (Maarten Krol) and NOAA-ESRL (Steve Montzka). In the revised manuscript we have expanded the text as indicated in our reply to the first comments, which discusses the methane lifetime.*

P9 l31 '…over the oceans G is the same as over the continents.' G is defined earlier (p2 l23) as 'gross OH formation'. I am unclear whether you mean G over the oceans as a whole compared to G over the continents as a whole, or if you mean per unit area. Obviously this makes a big difference.

*Reply: We have replaced the ambiguous formulation with the following sentence:*
*"At low latitudes G is much higher over continents than oceans, related to strong OH recycling, while at high latitudes longitudinal gradients are small, also between oceans and continents in the NH (Fig. 7)."*

P10 l7 '…S is also the same over the oceans and continents…' Same query as previous.

*Reply: This has been changed into:*
*"Consequently, on average S is also the same over oceans and continents,…"*

P10 l11 '…P declines steeply with solar radiation and water vapor.' Figure 7 show that P declines steeply with increasing altitude (ignoring the stratosphere) and latitude. Water vapor declines with increasing latitude and altitude (so that's OK). Solar radiation declines with increasing latitude, but increases with increasing altitude. So the relationship of P with solar radiation seems more complex than stated.

*Reply: We have reformulated into:*
*"Comparison of the middle and lower panels in Fig. 7 shows that spatial gradients of P and S can be rather different, e.g., towards high latitudes with P falling off with solar radiation and water vapor, while the latter also declines with altitude."*

P10 l30 'is subordinate to' -> is less than?

*Reply: changed into "less than"*

P11 l1-2 Isn't 'r' larger in the extra-tropics mainly just because P is small?

*Reply: We changed the sentence into (added "relatively"):*
*"Fig. 9 shows that r is relatively larger in the extra-tropics than in the tropics, and largest at high latitudes."*

P11 l2 The last sentence is true for the MBL but not the CBL, so it is incorrect for the BL as a whole.

*Reply: Has been corrected.*

P11 l4-13 Some clarification of what is exactly meant by the NOx, O3 and OVOC 'mechanisms' of OH recycling is needed. The earlier reaction equations and discussion is very good and useful, but I am not completely clear on which reactions make up each mechanism.

*Reply: In the revised manuscript we have more clearly defined the three recycling mechanisms, notably in the Introduction and by referring to Fig. 6. For example, referring to Fig. 6 we mention in the revised text:*
*"S is subdivided into contributions by the $NO_X$ mechanism (R7, blue), the $O_X$ mechanism (R11 and R12; green and yellow, respectively) and all OH recycling in VOC chemistry, the OVOC mechanism (red)."*

P11 l13 Do you mean from the FT to the BL (rather than 'transport in the FT')?

*Reply: We have changed the sentence into:*
*"Hence the $O_X$ mechanism depends on replenishment of $O_3$ through transport in the FT and subsequent mixing into the BL."*

P11 l17 'The complementarity of the three mechanisms is remarkable.' Is it? Don't they have to add up to 100% by definition? Figure 10 is certainly interesting, but I am not sure it is 'remarkable'. As suggested earlier, clearer definitions of the three mechanisms would help the discussion.

*Reply: We have deleted the word remarkable, and have more clearly defined the three mechanisms through reference to Fig. 6.*

P12 l4 Have you demonstrated in this paper that including MOM 'increases OH reactivity'? I can believe this is the case, but I don't think you present evidence of what the OH reactivity was in the model before you included MOM.

*Reply: In the revised manuscript we refer to Nölscher et al. (2016).*

P12 l13 I note your reference to 'measurement campaigns'. There is no comparison with observations in this paper, which seems like an oversight. Can you demonstrate that modelled OH is improved and compares well to reality?

*Reply: In the past our model simulated OH over the tropical forest was a factor of 2-3 too low compared to measurements, whereas with MOM the agreement has much improved. This will be discussed in detail by Taraborrelli et al. (in preparation), and has been partly discussed in Cabrera-*

*Perez et al. (2016). In the revised manuscript we have included the following text:*
*"Modeled OH concentrations over the tropical forest are about 1-2×10$^6$ molecules/cm$^3$, in agreement with OH measurements in South America and Southeast Asia (Kubistin et al., 2010; Pugh et al., 2010; Whalley et al., 2011)."*

P18 Table 1. The caption doesn't adequately describe the table – which contains fluxes for HOx primary production (O1D+H2O), recycling (NO+HO2, O3+HO2, photolysis reactions) and loss (H2O2 deposition). This table could be more comprehensive, and describe the full OH and HO2 budgets, i.e. include all the primary sources, OH to HO2 inter-conversions, and sinks (e.g., Derwent, 1996). The sources and sinks should balance (this is not obvious from the current table). If this were done, it could also clarify the definitions of the three mechanisms, as suggested earlier.

*Reply: We assume ref#2 refers to Fig. 1 in Derwent (1996), which is similarly structured to Fig. 2 in Lelieveld et al. (2002). In the revised manuscript we have completed Table 1 with sources and sinks of OH, to present a full budget (and checked that sources and sinks balance).*
*We changed the caption into:*
*"Global, annual mean tropospheric source and sink fluxes of OH (Tmol/yr). Sources and sinks are also specified for the boundary layer and free troposphere."*

P19 Figure 1 (and all zonal mean plots). It looks like surface pressures go up to 1000 hPa everywhere, but that can't be the case over Antarctica (etc.). Is the vertical scale really pressure?

*Reply: This is correct. Hence the lower levels over Antarctica in the zonal plots are white. Since we are presenting zonal averages some areas up to 1000 hPa occur almost everywhere, except Antarctica.*

P21 Figure 6. Related to my comments on Table 1 – I note the caption says 'Main' production terms of OH… Wouldn't a figure that shows all the OH sources be more useful?

*Reply: Considering the comprehensive chemistry scheme in MOM, the figure would become complicated in the VOC part of the pies. The aim of such a figure is to provide an overview of the main production terms, i.e., primary formation and the three main OH recycling mechanisms. For the more detailed description of the OVOC fluxes of OH we refer to Taraborrelli et al. (this will be the main topic of that paper). We have provided additional discussion on the three recycling mechanisms in Sect. 6, which fits to Fig. 6 and Table 1.*

P24 Figure 9. The text defines recycling efficiency as (S-P)/G. This allows it to take on negative values (where P>S). I find this a bit confusing, as efficiency normally refers to a number between 0-100%. The recycling probability (S/G) does just go from 0-100%. I'm not sure you need both quantities?

*Reply: In the revised manuscript we have removed the recycling efficiency (left panel of Fig. 9).*

P25 Figure 10. Again, clear definitions of the three mechanisms are needed.

*Reply: This has been remedied in the revised manuscript.*

Ref#3

This paper discusses the global OH atmospheric chemistry and analyzes the levels and chemical properties and the recycling of OH and HO2 using the global modelling system EMAC. The paper reads smoothly at the most part, providing a lot of information on the chemistry of OH, and an abundance of results both in the main manuscript and the supplementary material.

General Comments
The abstract should be more precise on what the main findings of this work are. The predecessor model as well as the previously assumed amounts of secondary sources should be specifically mentioned here.

*Reply: We have added the following to the abstract:*
*"By accounting for the complete breakdown products of higher VOCs, MOM is mass conserving, and calculates substantially higher OH reactivity from VOC oxidation compared to predecessor models. Whereas previously P and S were found to be of similar magnitude, the present work indicates that S may be twice as large, mostly due to OH recycling in the free troposphere. ... With an OH recycling probability of about 67%, global OH is buffered and not sensitive to perturbations by natural or anthropogenic emission changes."*

The terms "buffering" and "buffered" are used throughout the manuscript without proper definition given. Even after reading the entire manuscript it remains uncertain what the OH buffer actually is. The entire manuscript is based on the calculations made using an unpublished chemical mechanism (MOM) which is an update of a previous mechanism (MIM) using as reference a manuscript that is in preparation. Even though the full mechanism is included in the Supplementary material of the manuscript, a comparison of the model results using the updated mechanism to results of the previous mechanism and a more detailed comparison to measurements is needed. Also a better more complete budget analysis as well as a comparison and highlight of the differences between the two versions is clearly missing, especially since the authors give relative results such as "higher", or "compared to predecessor models".

*Reply: We have added the following definition of buffering to the introduction, and evaluate our results accordingly:*
*"Lelieveld et al. (2002) applied perturbation simulations of $NO_X$ and $CH_4$ to compute the impact on OH. This showed that at an OH recycling probability of 60% or higher, these perturbations have negligible influence on OH (their Fig. 6). Therefore, we argue that at r > 60% the atmospheric chemical system can be considered as buffered."*
*In the revised manuscript we have presented a more complete budget (Table 1) and compare results with previous work.*

The model description is rather short and feels incomplete. The EMAC modeling system is a complex system with a variety of options. The specific sub models used as well as the input used for the present study (i.e. emissions) should be clearly mentioned in the manuscript even if there is a small analysis in the supplementary material. The choice of RCP8.5 that suggests no further emission control also seems strange as it is often used to simulate the worst-case scenario. Even if in the year of interest (2013) the differences from the other scenarios are small, it still is an interesting choice and one that normally should be justified.

*Reply: In the revised manuscript we provide additional model description, based the model version published by Jöckel et al. (2010), with modifications that have been documented by Yoon and Pozzer (2014) and Cabrera-Perez et al. (2016), who also used MOM.*
*Page 17 of the supplement provides a detailed list of emission fluxes applied in the model. While the RCP8.5 scenario is a business-as-usual scenario for $CO_2$ emissions, it is more conservative for reactive trace gases and generally conceived as more realistic than the other scenarios. In Sect. 2 we have added a description of the RCP8.5 scenario and included additional model description and references.*

Finally I would suggest that a label is added by the colorbar of all figures, indicating the depicted property/substance and the units. This would make the interpretation of the figures quite easier.

*Reply: In the revised manuscript we have added labels to the figures.*

Specific Comments
P1, L10: ...may be significant... ! change to something more precise, or explain the reason that they might not be significant.
In all the reactions: Add the radical sign (dot) where necessary.

*Reply: We have added the following sentences:*
*"Whereas previously P and S were found to be of similar magnitude, the present work indicates that S may be twice as large, mostly due to OH recycling in the free troposphere. ... With an OH recycling probability of about 67%, global OH is buffered and not sensitive to perturbations by natural or anthropogenic emission changes."*
*While adding dots to radicals is generally good practice in chemistry, it is not common in atmospheric chemistry publications. One would have to do it with OH, but also other radicals, which can become problematic for larger molecules. We have added the following to the text:*
*"Note that the formal notation of hydroxyl is HO·, with one unpaired electron on the oxygen atom. For brevity we omit the dot and use the notation OH, and similarly for other radicals."*

*P3*, L25: R6 does not directly produce OH, hence a more clear explanation of how OH is produced is needed.

*Reply: We have added the complete list of reaction channels for R6, and changed*

$RO2 + HO2 \rightarrow ROOH + O2$ (R6)
*into*
$RO2 + HO2 \rightarrow ROOH + O2$    (R6a)
        $\rightarrow RO + O2 + OH$   (R6b)
        $\rightarrow ROH + O3$     (R6c)

*and on p.3 line 25 we refer to R6b.*

P3, L26: While in polluted air peroxy.... ! While in polluted air, peroxy...

*Reply: the comma does not belong here; has been changed*

P4, L15: By interconnected, do you mean coupled? If yes, the more used (and easier to understand) term should be used. If not, please give a definition of what an interconnected sub model is.

*Reply: replaced by coupled.*

P4, L27/28: Since only one year of results is presented (2013), why is a range of emitted quantities provided?

*Reply: We have performed a 4-year simulation and only show the last year. In the 4-year period the online calculated emissions vary somewhat. We have specified this in the revised manuscript.*

P5, L24/25 and elsewhere in the manuscript: Add the 105 term to the first number of all ranges.

*Reply: done*

P5, L27: Give the numbers calculated by Patra et al., since the discussion is based on them.

*Reply: Patra et al. conclude inter-hemispheric parity (is in the title of their publication).*

P7, L4: Reaction R1 of the manuscript should be referenced here.

*Reply: done*

P7, L7/8 and figure 2 caption: scaled down by a factor of 20.

*Reply: mentioned in the figure (units added)*

P7, L28/29: O3 from the stratosphere and O3 from photochemically... ! O3 from both the stratosphere and photochemically...

*Reply: has been changed*

P12, L15: The Physical-chemical tele-connections is here used without prior definition. Please give a clear definition.

*Reply: In the revised manuscript we refrained from using the work tele-connection.*

Figure 5: Add the OH reactivity zonal means (latitudinal) since the height distribution is mentioned during the discussion in section 5.

*Reply: done*

Figure 6: Enlarge the third panel of the figure since it is quite difficult to read the numbers in it. Also review the percentages given here since the numbers (as they are provided now) do not add up: e.g. for the OVOCs (red) the FT is 12% and the BL 19%. Multiplied by the 86% and 14% ratios respectively it gives a total of 13% (12.98) in the troposphere, where you present 12%. Maybe give the numbers with at least one decimal point so that the math comes out correct.

*Reply: We have enlarged the numbers in the third panel for readability and double-checked the numbers; slight inconsistencies may arise from truncation, which should not be a problem, but in the table the OH budget is closed. Additional decimal numbers would suggest a level of accuracy that does not do justice to the meaning of our results, which aims at conceptual understanding of OH recycling.*

[revised manuscript text omitted]

---

## Author Response (AR2)

In this revised manuscript, the authors present, in great details, their simulation results of OH, HO2, OH reactivity and OH recycling efficiency using a new chemical mechanism called "MOM". This is certainly an important topic and the manuscript is interesting, very detailed and is well written. However, I have some major concerns, which are:
1. The manuscript is aiming at informing the scientific community that the MOM solves the known biases in OH, especially, over low NOx, high isoprene regions (e.g., over the Amazonia), but without any experimental scientific evidences. Reaction rates and branching ratios can only be verified in smog chambers.

*Reply: We thank the reviewer for emphasizing the importance of experimental work and smog chambers. We agree, and for that reason have performed studies in our lab, to which we refer in Sect. 2 (e.g., Groß et al., 2014a,b), and also refer to work in other labs (e.g., Paulot et al., 2009; Crounse et al., 2012, 2013; Fuchs et al., 2014; Peeters et al., 2014). We have tested our isoprene degradation mechanism in the EUPHORE reaction chamber (Nölscher et al., 2014), leading to important improvements. Our group is one of few who have performed measurements over tropical forests, showing experimentally that OH recycling in isoprene chemistry over the Amazon is a prerequisite to explain observed OH concentrations, which has been confirmed by similar measurements in S-E Asia (Pugh et al., 2010). These experimental studies have been complemented by theoretical studies in our group by Vereecken and co-workers. Hence we do not agree that this work is without experimental evidence.*

2. The unverified new reactions and secondary OVOCs in the MOM is a fundamental issue in the article. The previous MIM1 and MIM2 (Taraborelli et al., 2009) are reduced mechanisms from well-known MCMv3.1, which are the ones that have been subject to validations in the mentioned EMAC literatures. As of MIM3 and MOM, there are no validation studies and/or sufficient evidence that these chemical mechanisms would hold compared to laboratory, smog champers experiments or field measurements or even other mechanisms, e.g., MCMv3.3. Further detailed validations are necessary before scientific conclusions can be made based on these, yet, theoretical schemes.

*Reply: Our isoprene mechanism has been a complement to the MCMv3.2 mechanism, as the latter was developed with a focus on anthropogenic emissions, and we tested the scheme experimentally in Nölscher et al. (2014). Unlike other isoprene oxidation mechanisms in use, the scheme in MOM has been tested. The critical reactions involve H-migration within oxygenated reaction products, which are not unverified, as indicated above. Our aromatics scheme has been tested against the MCMv3.3 mechanism (Cabrera-Perez et al., 2016). The latter reference is new and also used MOM; we have added this reference to the revised manuscript. Since it is not possible to test all >1,600 reactions in our scheme experimentally, we have included the entire list to the supplement, for scrutiny in the community. This follows the philosophy of the MCM, which similarly includes reactions that have not been verified experimentally. While we do not agree with the referee that there are no validation studies, we do agree that further validation is necessary. However, this can hardly be a prerequisite for our model study, which has the character of putting forward a theory that can be falsified by experiments.*

3. The information on MOM in Page 12, lines 12-21 is the most important part in the manuscript and I suggest that the authors move it to the very beginning of the discussion. With these infos, the authors are encouraged to include

some validation/comparison to understand the significance of these updates on the results. Also how these updates compares to the updates in the MIM3 (Taraborelli et al., 2012)?

*Reply: In Sect. 6 we explain the buffering mechanisms. By moving the paragraph on OVOC buffering to the beginning of the discussion the order of explanations would be interrupted. However, we agree that it would be helpful to have some of this information earlier in the paper. Therefore, we included the following sentence to p.4, l.17: "When the OH concentration is low, its formation is maintained by photo-dissociation of HPALD, while at high OH concentration its sink reaction with HPALD gains importance." We will follow the encouragement of the reviewer by publishing a follow-up paper by Taraborrelli et al., which will be submitted to ACP soon. Experimental evidence of the revised role of h-shifts, leading to HPALD, can be found in Nölscher et al. (2014).*

4. Why only MOM captures this higher OH recycling in the free troposphere? Recent measurement-modeling studies (e.g., Nicely et al., 2016 and Anderson et al., 2015) found that global models underestimate OH by 20-30% related to uncertainties in the underestimated NOx levels as well as HCHO in the upper troposphere. Thus, the "new" increased OH reactivity in the MOM, without accounting to these factors, may cause an excessive recycling and thus unnecessary higher production of OH. This is a primary concern given that MOM is yet not validated with measurements.

*Reply: We thank the reviewer for pointing to the interesting publications from the Anderson group. However, Anderson et al. (2016) did not address OH but rather $O_3$, suggesting that biomass burning, notably in topical Africa and SE Asia, is a dominant source. Both papers make use of aircraft measurements during the CONTRAST campaign, and data of $O_3$, CO, NO, HCHO, $H_2O$, $C_3H_8$, $CH_4$, $C_5H_8$, $CH_3COCH_3$, $CH_3OH$ and $CH_3CHO$ mixing ratios, and J(O1D) and J(NO2) are reported. OH and $HO_2$ radicals, nor tracers that could constrain biogenic VOC chemistry, were measured. The empirically based OH column ($OH^{COL}$) reported by by Nicely et al. (2016) was obtained by constraining a chemical box model. Subsequently, the empirical $OH^{COL}$ concentrations were compared with CTMs, indicating that these models underestimate $OH^{COL}$, while they also underestimate NOx. The final conclusion by Nicely et al. (2016) was that "our calculations do not support the prior suggestion of the existence of a tropospheric OH minimum in the tropical western Pacific, because during January–February 2014 observed levels of $O_3$ and NO were considerably larger than previously reported". We do not see how this interpretation of data over the tropical Pacific Ocean, as remote from biogenic VOC emissions as conceivable in the easterly trades, might indicate that MOM may lead to excessive recycling and unnecessary higher production of OH. Perhaps some of the model underestimated NO indicated by Nicely et al. (2016) is related to the release from reservoir species in which VOCs play a role. We will look into this possibility in future work.*

General comments:
Page 1, line 6: ozone photolysis is the primary OH source on global scale; on regional scale there are other important sources (e.g., Stone et al., 2012). Authors may correct the sentence such as: insert "is mainly: after "the former".

*Reply: done*

Page 1, line 8: Why did the authors use the EMAC model as "general circulation model" and not in the CTM mode (i.e., using prescribed meteorology), typically used for chemistry studies to isolate the dynamics effects?

*Reply: In this work EMAC, which is a chemistry-general circulation model, was indeed used in the CTM mode. We have highlighted this in the manuscript, referring to Deckert et al. (2011).*

Page 1, line 9: Perhaps the authors can shortly elaborate, why the MOM produces higher OH reactivity? e.g., higher concentration/number of VOC oxidation products..etc, why?

*Reply: The main reason why MOM predicts higher OH reactivity is the large number (43) of primarily emitted VOCs that are accounted for (see Figure below). Furthermore, the degradation of the oxidation products is continued to the final product $CO_2$, and the reactivity of the reaction intermediates can be large. Another reason for the increased OH reactivity modeled with MOM is due to the use of an updated Structure Activity Relationship (SAR) for estimating the rate constants for OH (Nölscher et al., 2015). This SAR will be detailed in the manuscript of Taraborrelli et al. (in preparation).*

CH$_4$  HCHO  HCOOH
CH$_3$ONO$_2$  CH$_3$OH

isoprene

MBO

benzene  toluene  styrene

xylenes  trimethylbenzenes

alpha-pinene  beta-pinene

ethyl benzene

higher aromatics

3-carene  sabinene  camphene

phenol

MVK  MACR

benzaldehyde

Experimental evidences?

*Reply: See replies above.*

Page 1, line 11:
The primary questions here are: Why only MOM captures this higher OH recycling in the free troposphere. Recent measurement-modeling studies (e.g., Nicely et al., 2016 and Anderson et al., 2015) found that global models underestimate OH by 20-30% related to uncertainties in the underestimated NOx levels as well as HCHO in the upper troposphere. Thus, the "artificially" increased OH reactivity in the MOM, without accounting to these factors, may cause an excessive recycling and thus unnecessary higher production of OH. This is a primary concern given that MOM is yet not validated with measurements.

*Reply: This repeats the remarks of comment 4 above. We have replied accordingly. We object to the tendentious remarks of "artificially increased OH reactivity" and "excessive recycling and unnecessary higher production of OH", which are unfounded.*

Page 1, line 13: If the authors mean by "OH is buffered" that OH is not sensitive to changes in VOC of OH precursor levels, why is that? If S (VOC+OH→HO2+NO→OH) is higher than primary OH productions (e.g., O(1D)+H2O=OH), it is because of the high VOC and NO load, and the atmosphere should then be sensitive to primary OH sources (the limiting factor). Thus any increase or decrease in the primary OH precursors should affect OH levels. This is important, since the inclusion of any additional sources (e.g., Nicely et al., 2016, Anderson et al., 2015) to the MOM will disturb the current budget, again, which is not experimentally verified.

*Reply: We apologize but do not understand this comment. Does the referee mean that the current budget is not experimentally verified? We would be eager to learn which budget has been experimentally verified.*

Page 1, line 14: OH primary formation (i.e, form O3 photolysis) is the primary source of OH, though much smaller than the secondary formation. Thus, OH primary formation is not "complementary" to the secondary formation, i.e., without OH primary formation, OH will be depleted with time (via HO2+O3, HO2+HO2, OH+NO2, ..etc). The authors are advised to revise the statement in line 14.

*Reply: We have added "primary and secondary" to l.14.*

Page 1, line 21: What about other primary sources of tropospheric OH, e.g., alkene ozonlolysis, (e.g., Stone et al., 2012)?

*Reply: As shown in the supplement, these reactions are included in our mechanism. In some cases such reactions could contribute to boundary layer OH formation at night, while they do not play a significant role in the global OH budget, as discussed in our manuscript. Stone et al. (2012) discuss these reactions primarily because they can lead to spurious OH formation in instruments that apply the LIF-FAGE technique.*

Page 2, line 17: The statement "In air that is directly influenced by pollution emissions S is largely controlled by nitrogen oxides (NO+NO2 =NOx )" is not clear. In highly polluted urban conditions, ozone photochemical formation, which is the secondary product of S, is typically VOC limited, since NO emissions is too high compared to HO2. Authors are advised to elaborate here, what is the source of this info's, examples? Properly the authors meant "high isoprene emissions"?

*Reply: We believe that this sentence is clear and that it is in line with the well-known fact that in polluted air OH recycling is dominated by the reaction NO+HO2 (R7). It has nothing to do with isoprene emissions.*

Page 2, line 25: What is "self-limiting"?? At high NOx levels (polluted conditions), NO2 is a permanent sink of OH.

*Reply: It means that reaction R7 recycles OH, but reaction R10 becomes such a large sink of OH that it limits the net OH production. This is well known and elaborated in textbooks.*

Page 3, line 26: how this new MOM would compare to MCMv3.3.1 (Jenkin et al., 2015) or to measurements?

*Reply: The isoprene mechanisms in MOM and MCMv3.3 have much in common, as will be elaborated by Taraborrelli et al. (in preparation). Perhaps the referee can check with the group of Jenkin to verify about experimental validation against smog chamber experiments.*

Page 3, lines 31-33: Why the authors decided to use EMAC GCM as opposed to the CTM modes, typically used in atmospheric chemistry studies. How the authors would account for dynamics effects on their results in the case of GCM?

*Reply: This question has been answered above.*

Page 4, lines 20-25: The MIM1 and MIM2 are considered as reduced mechanisms from MCM (Taraborelli et al., 2009), which are the ones that have been subject to validations in the mentioned EMAC literatures. As of MIM3 and MOM, there are no validation studies and/or sufficient evidence that these theoretical mechanisms would hold compared to measurements. Further validations are needed before further scientific conclusions can be made based on these pure theoretical studies.

*Reply: This comment has also been replied to in the above replies.*

Page 5, line 10, 13: Since the simulation are performed for the year 2013, not any further, why the authors decided to use the RCP8.5, not, for example, a historical emissions scenarios, which should be available by the year 2016?

*Reply: Such scenarios are not available, and are not fundamentally important for the present study.*

Page 7, lines 23-29: The authors should mention that these OH distributions are based on "annual mean" and that seasonal OH distributions are different. For example, in the NH extra-tropics, OH in the MBL is not equal to that in the CBL, during July (Figure S1), otherwise the discussion is misleading. Authors are also advised to show the seasonal distribution instead, as in Figure 2 for nighttime OH.

*Reply: These distributions do not refer to annual mean, but to geographical mean in the MBL and CBL, as indicated in the text. The seasonal dependence is not relevant for this discussion.*

Page 7, lines 17-18: the authors mentioned "partly", what other sources could be, e.g., higher O3 photolysis?

*Reply: The point here is that the enhanced OH is related to the convective transport of VOC emissions and NOx from lightning. To express this more clearly we start the sentence with "The relatively high OH.." and have dropped the word "partly".*

Page 7, lines 30-32: So, here the authors mentioned the alkenes ozonolysis, therefore, they also need to mention it along with other OH sources in the introduction.

*Reply: As indicated above, this is not a major OH source. Instead of mentioning it in the introduction, it suffices that it is discussed in Sect. 3.*

Page 8, lin15-16: The statement is not clear. If sources of OH were high, OH would not have been depleted while HO2 is high. Would it be clearer to say that the reason for high HO2 but low OH is the low NOx condition, under which HO2 recycling efficiency is too low?. If OH is high because of the new introduced high OH recycling in MOM, then the authors should use this occasion to discuss why their new chemical mechanism works here.

*Reply: The OH sources as well as sinks are strong, so that overall OH is not very high, but since OH is converted into HO2, total HOx can still be high. Why would the HO2 recycling efficiency be too low? We do not understand this. In MOM the OH is higher over the forest than most other models that do not account for OH recycling in isoprene chemistry. This was shown by Taraborrelli et al. (2012) and Nölscher et al. (2014) based on MIM3. This version of the isoprene chemistry is also part of MOM.*

Page 8, line 17: could the authors provide reference for the Strong NOx emissions from petroleum industry in the Gulf of Mexico?

*Reply: We have added the reference Ren et al. (2013) for the Mexican and Lelieveld et al. (2009) for the Persian Gulf.*

Page 9, lines 21-23: This is the first statement that mentions comparison with previous mechanism or models, very briefly though!

*Reply: ok*

Page 9, lines 26: Add the standard deviation, i.e. Error of the mean lifetime of methane.

*Reply: A standard deviation here would not give additional information, as only the year-to-year standard deviation can be calculated (i.e., an indication of OH inter-annual variability). Hence it would not add information on the range of the methane lifetime. It may also confuse the reader as later in the text the methane lifetime from literature is mentioned, together with the standard deviation from the model ensemble, which is a different metric.*

Page 9, line 31, which observation-derived estimates, reference(s)?. As of Prinn et al. (2005), the mean is 10.2 years.

*Reply: We refer to Naik et al. (2013) who included Prinn et al. (2005) and others for their estimate of "observation-based estimates". Our group has also been involved in such estimates, expressing caution with the interpretation (see Krol and Lelieveld, 2003; Montzka et al., 2011).*

Page 10, lines 7-8: Again here, the authors very briefly mention why MOM is different from other models, How the "MOM mechanism more efficiently recycles OH than other VOC chemistry schemes applied in global models"?, How reasonable is this approach, compared to laboratory and field measurements of these enhanced recycling reaction??

*Reply: Why does the referee keep repeating the same comment? We explain this in our manuscript, showing that the account of higher generation reaction products in VOC chemistry (often truncated or simplified in models) leads to higher and more realistic OH reactivity and OH recycling. Our isoprene oxidation scheme is the first (and presently the only one) that was tested in the EUPHORE reaction chamber (Nölscher et al., 2014).*

Page 10, lines 20-30: Again, it is difficult to conclude a scientific values from these numbers without comparison and contrasting with measurements, previous mechanisms (e.g., MIM2) or MCMv3.1, especially that the authors claims that this is a new advanced MOM?

*Reply: see above.*

Page 12, lines 12-21 to Page 13, line 18: Here we go; actually this part is the most important part in the manuscript. Ok, so now, the authors need to add some validation/comparison to understand the relevance of these updates on the results. Also how these updates compares to the updates in the MIM3 (Traborelli et al., 2012)?

*Reply: see above*

Fig. 9: correct the caption.

*Reply: done*